# Novel RNA and DNA strand exchange activity of the PALB2 DNA binding domain and its critical role for DNA repair in cells

Jaigeeth Deveryshetty[1], Thibaut Peterlini[2], Mikhail Ryzhikov[1†], Nadine Brahiti[2], Graham Dellaire[3], Jean-Yves Masson[2], Sergey Korolev[1]*

[1]Edward A Doisy Department of Biochemistry and Molecular Biology, Saint Louis University School of Medicine, Saint Louis, United States; [2]Genome Stability Laboratory, CHU de Québec-Université Laval, Oncology Division, Laval University Cancer Research Center, Québec City, Canada; [3]Department of Pathology, Dalhousie University, Halifax, Canada

**Abstract** BReast Cancer Associated proteins 1 and 2 (BRCA1, −2) and Partner and Localizer of BRCA2 (PALB2) protein are tumour suppressors linked to a spectrum of malignancies, including breast cancer and Fanconi anemia. PALB2 coordinates functions of BRCA1 and BRCA2 during homology-directed repair (HDR) and interacts with several chromatin proteins. In addition to protein scaffold function, PALB2 binds DNA. The functional role of this interaction is poorly understood. We identified a major DNA-binding site of PALB2, mutations in which reduce RAD51 foci formation and the overall HDR efficiency in cells by 50%. PALB2 N-terminal DNA-binding domain (N-DBD) stimulates the function of RAD51 recombinase. Surprisingly, it possesses the strand exchange activity without RAD51. Moreover, N-DBD stimulates the inverse strand exchange and can use DNA and RNA substrates. Our data reveal a versatile DNA interaction property of PALB2 and demonstrate a critical role of PALB2 DNA binding for chromosome repair in cells.
DOI: https://doi.org/10.7554/eLife.44063.001

*For correspondence:
korolevs@slu.edu

Present address: [†]John T Milliken Department of Medicine, Division of Pulmonology and Critical Care Medicine, Washington University School of Medicine, Saint Louis, United States

Competing interests: The authors declare that no competing interests exist.

## Introduction

Breast cancer associated proteins 1 and 2 (BRCA1, −2) regulate an efficient non-mutagenic pathway of chromosome break repair and are described as guardians of chromosomal integrity (*Venkitaraman, 2014*). They initiate RAD51-mediated homologous recombination (HR) (*Davies et al., 2001*; *Moynahan et al., 2001*; *Sharan et al., 1997*; *Venkitaraman, 2000*) and facilitate restart of stalled replication (*Badie et al., 2010*; *Lomonosov et al., 2003*; *Schlacher et al., 2011*). BRCA2 belongs to a ubiquitous family of Recombination Mediator Proteins (RMPs), which stimulate formation of recombinase filament on single-stranded (ss) DNA protected by ssDNA-binding proteins, like SSB and RPA (*Beernink and Morrical, 1999*; *Cox, 2007*; *Kowalczykowski, 2005*). The Partner and Localizer of BRCA2 (PALB2) protein was discovered as a protein forming a complex with BRCA2 and regulating BRCA2 activity (*Xia et al., 2006*). Like BRCA proteins, PALB2 is an essential mammalian protein linked to a similar spectrum of cancers and Fanconi anemia (*Ducy et al., 2019*; *Pauty et al., 2014*; *Xia et al., 2007*). The PALB2 C-terminal WD40 domain interacts with BRCA2 (*Oliver et al., 2009*; *Xia et al., 2006*) while the N-terminus forms a complex with BRCA1 (*Zhang et al., 2009a*; *Zhang et al., 2009b*). The latter localizes at double-stranded DNA break (DSB) sites at earlier stage of repair, inhibiting an alternative pathway of non-homologous end joining and initiating homology-directed repair (HDR) through recruitment of PALB2/BRCA2/RAD51 (*Prakash et al., 2015*).

**eLife digest** DNA in a cell is under constant stress from environmental factors, such as ultraviolet light, or from damage caused by the replication process. These sources of stress can cause breaks in the genome, which if left unrepaired can lead to cancer or cell death. One of the most accurate ways to repair a broken fragment of DNA is through recombination – whereby an undamaged copy of the sequence is located in another DNA molecule and used as a template to replace the missing fragment.

DNA recombination is regulated by more than a dozen proteins that help recruit the enzyme RAD51 to sites of DNA damage, and trigger its search for complementary sequences of DNA. A molecule known as PALB2 binds to these DNA repair proteins and coordinates their activity. If PALB2, or these other proteins become mutated, this can increase the risk cancerous growths in various tissues, including the breasts and ovaries. Having a better understanding of how this group of proteins control the repair process could therefore improve prognosis and advance cancer treatments.

Now, Deveryshetty et al. have discovered a new and unexpected role for PALB2 within the recombination pathway. As well as binding to other repair proteins, PALB2 interacts directly with DNA, and this interaction was found to be an important part of the repair process. Even in the absence of RAD51, PALB2 was still able to recombine short fragments of DNA sequence. PALB2 achieves this by initiating recombination using single strands of DNA or a DNA-like molecule known as RNA. This latter property may be particularly important if the molecular machines needed to replicate DNA and synthesize RNA collide on the same DNA molecule.

This new role for PALB2 could lead to the discovery of other DNA repair mechanisms, and could be used to predict which PALB2 mutations are more likely to cause cancer. Patients who are at greater risk of cancer could then be treated with more advanced therapies, in order to increase their chances of recovery.

DOI: https://doi.org/10.7554/eLife.44063.002

PALB2 is often described as the hub for a network of tumor suppressors (*Park et al., 2014b*; *Sy et al., 2009b*). In addition to BRCA1 and −2 interactions, it contains a chromatin-association motif (ChAM) that interacts with histones H3 and H2B (*Bleuyard et al., 2012*). PALB2 binds MRG15 protein, a component of histone acetyltransferase-deacetylase complexes (*Hayakawa et al., 2010*; *Sy et al., 2009a*); RAD51 and its paralogs RAD51C, RAD51AP1 and XRCC3 (*Dray et al., 2010*; *Park et al., 2014a*); translesion DNA polymerase η (Polη) during recombination-associated DNA synthesis (*Buisson et al., 2014*); an oxidative stress response protein KEAP1 (*Ma et al., 2012*); and RNF168 ubiquitin ligase (*Luijsterburg et al., 2017*). PALB2 is ubiquitinylated in G1 phase of the cell cycle by KEAP1 and CUL3, leading to its degradation, and, thereby, restraining its activity in S/G2 (*Orthwein et al., 2015*).

Furthermore, PALB2 is an RMP itself as it promotes the assembly of RAD51-ssDNA presynaptic nucleofilaments in the absence of BRCA2 in vitro (*Buisson et al., 2010*; *Dray et al., 2010*). PALB2 recruits Polη to DSB sites and stimulates recombination-associated DNA synthesis by Polη (*Buisson et al., 2014*).

Apart from protein-protein interaction domains, BRCA1, −2 and PALB2 proteins also contain DNA binding domains (DBDs), the function of which is poorly understood (*Buisson et al., 2010*; *Dray et al., 2010*; *Paull et al., 2001*; *Pellegrini et al., 2002*). Recently, studies of BRCA1/BARD1 complex interaction with DNA and RAD51 led to the discovery of the BRCA1/BARD1 role in RAD51-mediated strand invasion and D-loop formation (*Zhao et al., 2017*). Most missense mutations in the BRCA2 DBD are pathogenic (*Guidugli et al., 2013*; *Wu et al., 2005*). Disruption of DNA binding in a BRCA2 truncation variant lacking the PALB2-binding motif leads to a significant HDR reduction (*Siaud et al., 2011*). However, deletion of the BRCA2 DBD has a negligible effect when interaction with PALB2 is preserved, highlighting the functional importance of the previously reported DNA-binding property of PALB2 (*Buisson et al., 2010*; *Dray et al., 2010*). Two truncation fragments of PALB2, T1 (residues 1–200) and T3 (residues 372–561), bind DNA (*Buisson et al., 2010*). While the precise mechanism of DNA interaction and its functional role remain unknown, several reports

indirectly support its importance in DNA repair. For example, PALB2 truncation of 500 amino acids situated between the BRCA1 and BRCA2 binding motifs and containing both DBDs does not support BRCA2 and RAD51 foci formation in cells during DNA damage (*Sy et al., 2009b*). Since both the BRCA1-binding N-terminal and the BRCA2-interacting WD40 C-terminal domains are retained in this mutant, the results points to the potential importance of DBDs in PALB2 function. The importance of DNA binding was demonstrated for other RMPs, including bacterial RecFOR (*Korolev, 2017*; *Morimatsu and Kowalczykowski, 2003*; *Ryzhikov et al., 2014*; *Sakai and Cox, 2009*; *Umezu et al., 1993*) and eukaryotic RAD52 (*Arai et al., 2011*; *Seong et al., 2008*).

In the current study, we identified a major DBD of PALB2 (N-DBD) and specific amino acids involved in DNA binding. Mutations of only four amino acids significantly reduce RAD51 foci formation and the efficiency of HDR in a model cell system. PALB2 N-DBD by itself stimulates RAD51 strand exchange reactions. Surprisingly, we found that the PALB2 N-DBD supports both forward and inverse strand exchange even in the absence of RAD51 and can use RNA as a substrate. Altogether, our data reveal a novel activity of PALB2 and highlight the importance of PALB2 DNA binding in chromosome maintenance.

## Results

### The DNA-binding mechanism of PALB2 and its function in DNA repair

#### The major DNA-binding site of PALB2 is localized in the N-terminal domain (N-DBD)

A functional significance of PALB2 interaction with DNA was suggested by several studies described above, including a deletion encompassing both DBDs, the T1 (1–200) and T3 (372–561) fragments (*Sy et al., 2009b*). However, this deletion also removed several protein interaction sites. In two published studies the T1 and T3 fragments displayed different DNA binding activities measured by gel shift assays, but the cause remains unclear (*Buisson et al., 2010*; *Dray et al., 2010*). We used a quantitative fluorescence polarization (FP) method to investigate interaction of PALB2 fragments with ss- and dsDNA oligonucleotides of different lengths. Both T1 and T3 fragments and the fragment consisting of amino acids 1–573 (PB2-573 in text), which includes both the T1 and T3, were cloned and purified (*Figure 1—figure supplement 1*). T1 fragment alone interacts with all tested substrates with similar affinities as of PB2-573, while the T3 fragment has significantly lower affinity for DNA (*Figure 1*). For example, the apparent equilibrium dissociation constant of T1 binding to ss49 $K_{d(T1/ss49)}$ is 4.0 ± 1.3 nM, of 573 fragment $K_{d(573/ss49)}$ is 4.8 ± 0.4 nM, while that of the interaction of T3 with ss49 $K_{d(T3/ss49)}$ is 484 ± 80 nM, which is likely an overestimation due to weak binding. Hill coefficient values are close to one, suggesting non-cooperative interactions. The only exception is T1 binding to 49 bp dsDNA, however, the binding of PB2-573 to the same substrate is not cooperative. The only difference between T1 and PB2-573 was observed at an elevated salt concentration of 250 mM NaCl, where the PB2-573 fragment retained partial DNA binding activity (*Figure 1—figure supplement 2*). In both cases, interactions were inhibited by in 500 mM NaCl. The T1 fragment will be referred to as N-DBD in the text below. Interestingly, N-DBD binds long ssDNA substrates with significantly higher affinity than short ones with $K_{d(T1/ss20)}$ = 80 ± 8.5 nM for 20 nt ssDNA versus $K_{d(T1/ss49)}$ = 4.0 ± 1.3 nM for ss49. This suggests an interaction with ssDNA through multiple binding sites, potentially formed by the PALB2 oligomerization, as previously described (*Buisson and Masson, 2012*; *Sy et al., 2009c*), or through interaction with multiple binding sites within a monomer (see below). Interaction with dsDNA was length-independent, suggesting that more rigid dsDNA interacts with a single site.

#### Identification of DNA-binding residues

Since PALB2 DNA binding is salt-dependent (*Figure 1—figure supplement 2*), we performed an alanine scanning mutagenesis of several clusters of positively charged amino acids to identify the DNA binding site in the N-DBD (*Figure 2—figure supplement 1*). The main DNA-binding cluster is formed by amino acids R146, R147, K148, and K149. Alanine substitution at these residues reduced binding affinity to ss49 by two orders of magnitude with a change in an apparent $K_d$ from 4.0 ± 1.3 nM to 316 ± 59 nM in the case of N-DBD and from 4.8 ± 0.4 nM to 187 ± 55 nM in the case of PB2-573 (*Figure 2*). DNA binding was moderately affected by mutations of two other clusters, including

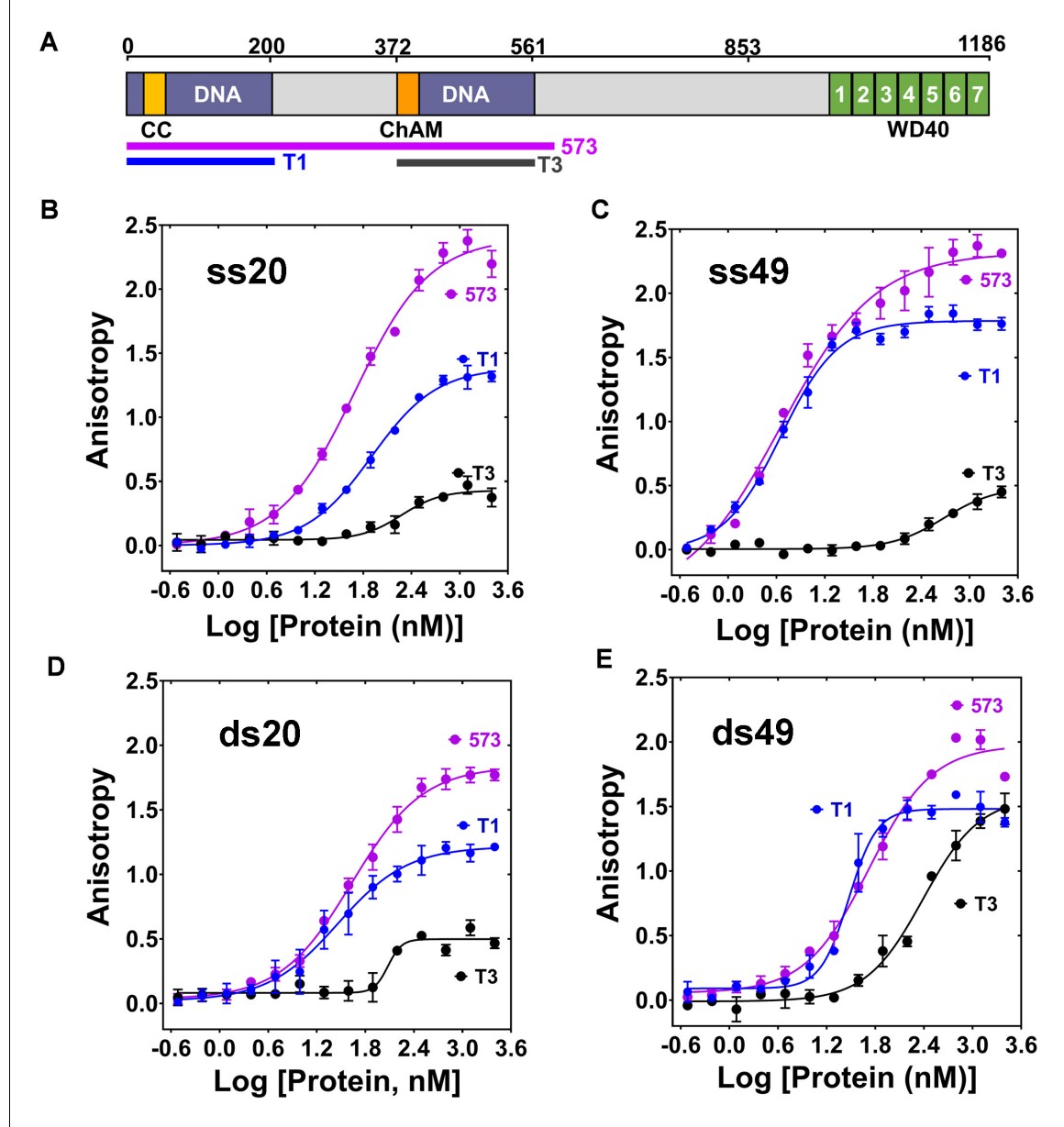

**Figure 1.** Interaction of T1, T3 and PB2-573 fragments with ss- and dsDNA. (A) Domain structure of PALB2. The PALB2 truncations used in the present study are shown below by magenta, blue and dark grey lines. (B–E) Equilibrium binding of PALB2 fragments, including T1 (blue), T3 (dark grey) and PB2-573 (magenta), to 20 nt ssDNA (ss20) (B), 49 nt ssDNA (ss49) (C), 20 bp dsDNA (ds20) (D), and 49 bp dsDNA (ds49) (E) monitored by fluorescence anisotropy of FAM-labelled ssDNA (5 nM). Each data point is an average of six readings from two different experiments. Reactions were performed in assay buffer with 20 mM Tris-acetate pH 7.0, 100 mM NaCl, 5% glycerol, 10% DMSO in a 40 μL reaction volume.

DOI: https://doi.org/10.7554/eLife.44063.003

The following source data and figure supplements are available for figure 1:

**Source data 1.** Table with Hill coefficients (n) and equilibrium dissociation constants ($K_d$, nM) values for graphs in *Figure 1B–E*.
DOI: https://doi.org/10.7554/eLife.44063.004

**Figure supplement 1.** SDS PAGE analysis of purified proteins used in this study: (A) T1 and T1 146AAAA mutant, (B) T3, (C) PB2-573 and PB2-573 146AAAA, (D) RAD51, and (E) RPA.
DOI: https://doi.org/10.7554/eLife.44063.005

**Figure supplement 2.** Effect of increasing salt concentration on PALB2 fragments binding to DNA.
DOI: https://doi.org/10.7554/eLife.44063.006

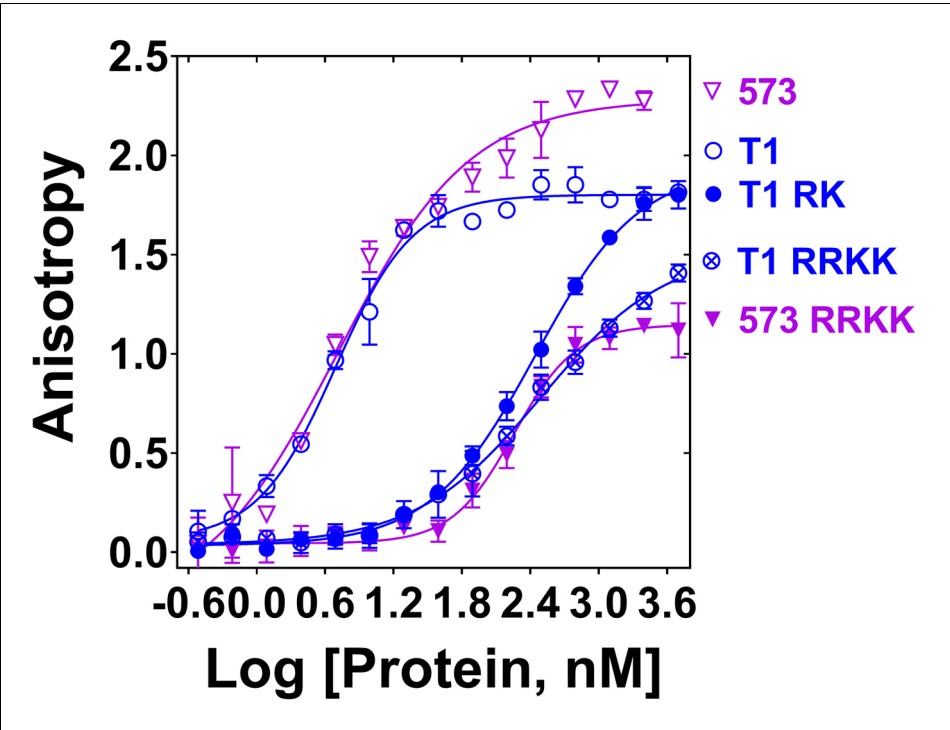

**Figure 2.** Mutation of DNA binding residues. Isotherms of fluorescence anisotropy of FAM-ss49 (5 nM) titrated by PALB2 T1 (blue, open circles) and PB2-573 (magenta, open triangles) fragments and their mutants: T1 146-RK/AA (filled blue circles), T1 146-RRKK/AAAA (crossed open blue circles), and 573 146-RRKK/AAAA (filled magenta triangles) under conditions identical to those in *Figure 1*.

DOI: https://doi.org/10.7554/eLife.44063.007

The following source data and figure supplements are available for figure 2:

**Source data 1.** Table with Hill coefficients (n) and equilibrium dissociation constants ($K_d$, nM) values.
DOI: https://doi.org/10.7554/eLife.44063.008

**Figure supplement 1.** Amino acid sequence alignment of PALB2 T1 from different organisms with residues colour-coded accordingly to polarity, with mutated residues identified by red boxes, and with the secondary structure elements depicted at the bottom of alignment in cartoon representation as predicted by the Phyre server.
DOI: https://doi.org/10.7554/eLife.44063.009

**Figure supplement 2.** DNA binding of T1 mutants.
DOI: https://doi.org/10.7554/eLife.44063.010

K45A/K50A, for which the $K_d$ was increased to 28 ± 5.2 nM (*Figure 2—figure supplement 2*), and the triple mutant R170A/K174A/R175A with similar change in $K_d$. From these experiments, we concluded that the main DNA binding site is formed by residues 146–149, with a potential minor contribution from other basic amino acids of the N-DBD.

## Impairment of DNA repair in cells with the PALB2 DNA-binding mutant

The mutations described above were used to separate the DNA-binding function from other macromolecular interactions of PALB2 during DNA repair in HeLa cells. Positively charged residues 146–149 were mutated to alanines in the full-length PALB2 protein and the effect of these mutations was measured in two assays. First, we evaluated RAD51 foci formation in cells after gamma irradiation (*Figure 3A*). Endogenous PALB2 was depleted by siRNA and cells were transformed with either wild type PALB2 or the DNA-binding site mutant (*Figure 3C*). PALB2 depletion leads to a severe defect in RAD51 foci formation. WT PALB2 restores RAD51 foci formation, while the DNA-binding site PALB2 mutant restores only ~50% of RAD51 foci formation. Similar effect was observed for the time frames between 1 to 8 hr after DNA damage (*Figure 3—figure supplement 1*). Therefore,

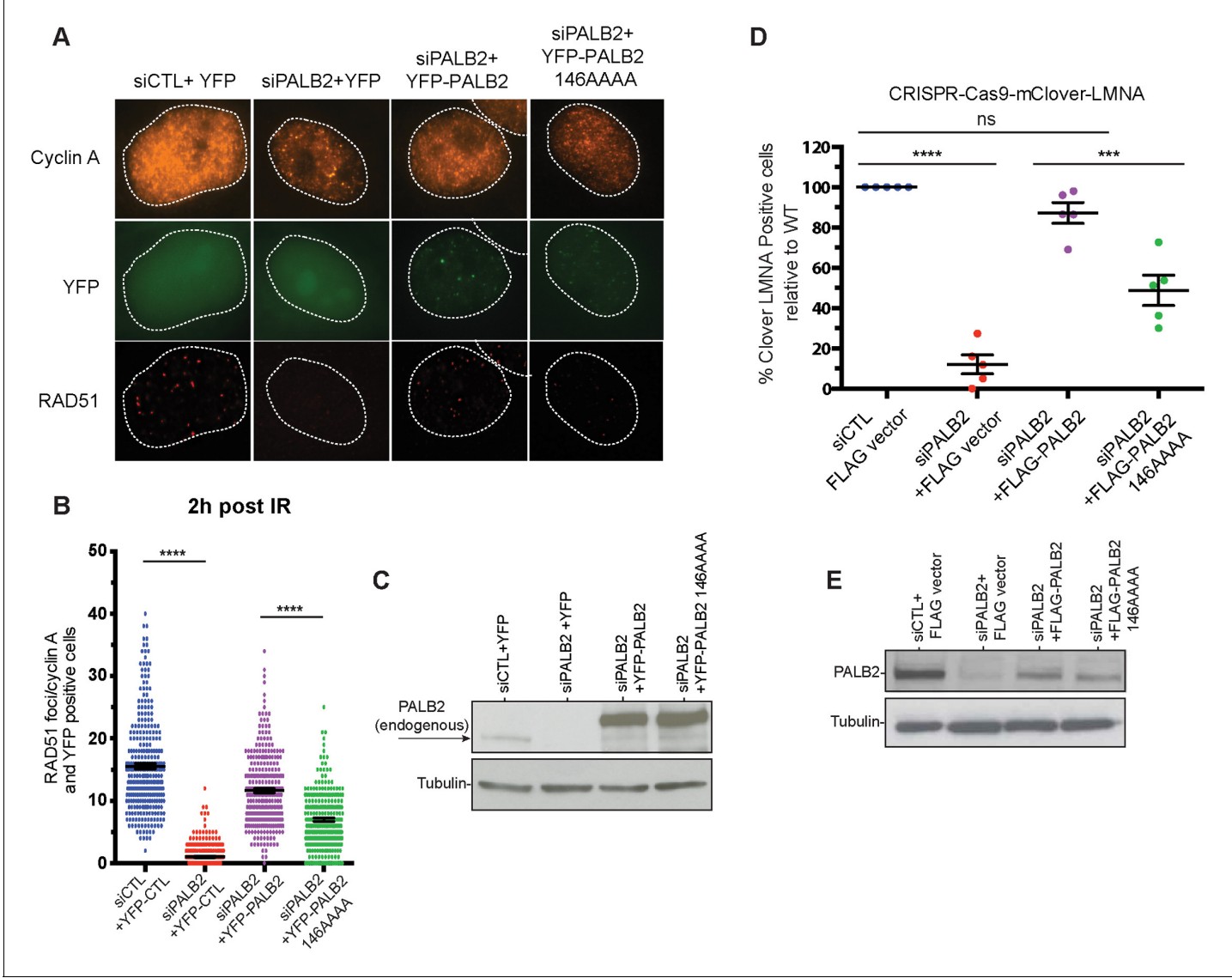

**Figure 3.** Effect of a PALB2 DNA-binding site mutation on homologous recombination. (**A**) Representative immunofluorescence images of RAD51 foci in PALB2 knockdown HeLa cells complemented with the indicated YFP construct and synchronized in S/G2 by double thymidine block, as determined by cyclin A staining. (**B**) RAD51 foci quantification in control siRNA (blue), siPALB2 (red) and with siPALB2 with subsequent complementation by siRNA-resistant constructs YFP-PALB2 (magenta) and 146AAAA DNA-binding site mutant PALB2 (green) at 2 hr after irradiation. (**C**) Western blotting of the samples shown in (**B**) to monitor knockdown and complementation efficiency. (**D**) Quantification of the gene-targeting efficiency of siRNA PALB2 cells complemented with wild-type and 146AAAA siRNA-resistant constructs mClover positive/iRFP cells. (**E**) Western blotting of the samples shown in (**D**) to monitor knockdown and complementation efficiency. ***p<0.001 and ****p<0.0001.

DOI: https://doi.org/10.7554/eLife.44063.011

The following figure supplements are available for figure 3:

**Figure supplement 1.** Time dependence of RAD51 foci formation after DNA damage in the experiment described in *Figure 3*.
DOI: https://doi.org/10.7554/eLife.44063.012

**Figure supplement 2.** Representative images and schematic representation of CRISPR-Cas9/mClover-LMNA1 mediated HR assay.
DOI: https://doi.org/10.7554/eLife.44063.013

**Figure supplement 3.** YFP-PALB2 and YFP-PALB2 146AAAA binds RAD51 equally.
DOI: https://doi.org/10.7554/eLife.44063.014

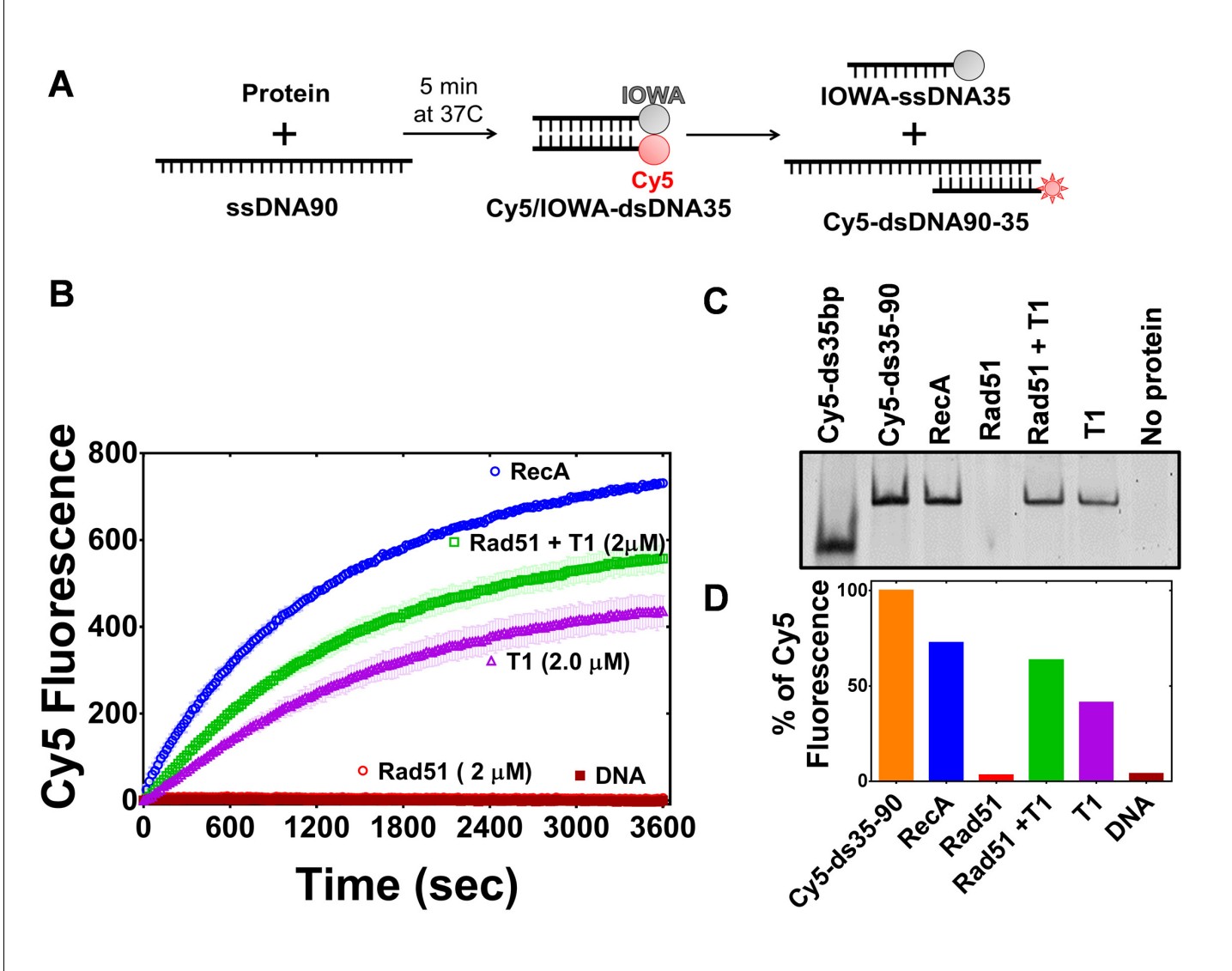

**Figure 4.** PALB2 promotes strand exchange between homologous DNA substrates. (**A**) Schematic representation of the strand exchange activity assay. ss90 (120 nM) was incubated with RecA (2 µM) or RAD51 (2 µM) for 5', then with PALB2 fragment (2 µM) for 5', then dsDNA (100 nM) was added and the Cy5 fluorescence was measured on a plate reader using 635 nm excitation and 680 nm emission wave-lengths. (**B**) Continuously measured Cy5 fluorescence after initiating reactions with RecA (blue), RAD51 (red), PALB2 N-DBD (magenta), RAD51 and PALB2 N-DBD (green), and without proteins (dark red). Each point is an average of three measurements. (**C**) Reaction products from (**B**) were deproteinized and separated on a native PAGE gel. Control in lane one represent Cy5-labelled dsDNA without IOWA quencher and in lane 2 Cy5-labelled ssDNA annealed with ss90. (**D**) Percentage of Cy5 fluorescence of the final reaction in (**B**).

DOI: https://doi.org/10.7554/eLife.44063.015

The following figure supplements are available for figure 4:

**Figure supplement 1.** RAD51 activity under optimized conditions.
DOI: https://doi.org/10.7554/eLife.44063.016
**Figure supplement 2.** PALB2 strand exchange activity with dsDNA with alternative fluorophores.
DOI: https://doi.org/10.7554/eLife.44063.017
**Figure supplement 3.** PALB2 strand exchange activity with longer dsDNA.
DOI: https://doi.org/10.7554/eLife.44063.018
**Figure supplement 4.** PALB2 N-DBD cannot form D-loops with supercoiled dsDNA plasmid.
DOI: https://doi.org/10.7554/eLife.44063.019

mutagenesis of only four positively charged residues in PALB2 has a major effect on efficiency of RAD51 recruitment to DNA damage sites.

Similarly, we tested the role of PALB2 interaction with DNA for the efficiency of HDR in U2OS cells using a novel LMNA-Clover based assay, where DNA breaks at a specific gene are introduced by the CRISPR-Cas9 system (*Figure 3—figure supplement 2*) (*Buisson et al., 2017*). As in case of RAD51 foci formation, complementation of PALB2-depleted cells with the DNA-binding PALB2 mutant restores only 50% of HDR efficiency, in contrast to WT PALB2, which restores more than 90% of activity (*Figure 3D*). The N-DBD fragment also possesses a RAD51 interaction site, which can be potentially affected by mutations. We verified that the 146AAAA mutant interacts with RAD51 similarly to wild type PALB2 (*Figure 3—figure supplement 3*). Therefore, the cellular effect of the 146AAAA mutant is defined exclusively by the inability of PALB2 to interact with DNA. Altogether, these studies show that PALB2 DNA binding plays a critical role in HR and DNA repair in vivo.

## PALB2 promotes DNA and RNA strand exchange

### PALB2 stimulates RAD51-mediated strand exchange and promotes a similar reaction without RAD51

PALB2 stimulates RAD51 filament formation even in the absence of BRCA2 (*Buisson et al., 2010*). Here, we investigated the ability of the PALB2 N-DBD to stimulate the strand exchange activity of RAD51 using a fluorescence-based strand exchange assay similar to the one previously published (*Figure 4A*) (*Jensen et al., 2010*; *Ryzhikov et al., 2014*). Under solution conditions used in the DNA-binding assays in *Figure 1* and even with reduced NaCl concentration, RAD51 displayed a low activity, in contrast to *E. coli* RecA (*Figure 4*). RAD51 activity was stimulated by addition of 5 mM $CaCl_2$ (*Figure 4—figure supplement 1*). RMPs stimulate recombinase activity even at unfavourable solution conditions, such as in the case of Rad52 (*Krejci et al., 2002*; *New et al., 1998*), BRCA2 (*Jensen et al., 2010*)(*Liu et al., 2010*; *Thorslund et al., 2010*) and the Hop2-Mnd1 complex (*Chi et al., 2007*). Similar to the previously published finding that the full length PALB2 stimulates RAD51 function (*Buisson et al., 2010*; *Dray et al., 2010*), we found that the PALB2 N-DBD alone stimulates RAD51-mediated strand exchange (*Figure 4*).

Surprisingly, the N-DBD promotes strand exchange at a comparable rate even without RAD51. Reaction products were further analysed by EMSA gel shift to rule out an artefact of protein-specific fluorophore quenching (*Figure 4C*). The results were confirmed using DNA with different fluorescent labels (*Figure 4—figure supplement 2*). The strand exchange activity of N-DBD was even more efficient with longer dsDNA substrates (*Figure 4—figure supplement 3*). The reaction does not require ATP, resembling those promoted by RAD52 (*Mazina et al., 2017*). The strand exchange activity was not observed by full length PALB2 in previous publications (*Buisson et al., 2010*; *Dray et al., 2010*). This difference could be due to significantly lower concentrations of PALB2 used in the referred studies as well as different experimental conditions. Currently, the low solubility of the full-length protein prevents recapitulation of assays in *Figures 5* and *6* at comparable concentrations.

Since N-DBD stimulates a similar reaction on its own, it is unclear whether the N-DBD fragment stimulates RAD51 activity or if the two proteins function independently. In a limited titration experiment shown in *Figure 5*, the efficiency of the strand exchange increases proportionally to N-DBD concentration. In the presence of 2 µM of RAD51, the activity at T1 concentration of 0.5 µM increases four times and the maximum rate of strand exchange is reached at 1 µM of N-DBD. These data suggest a synergistic effect of two proteins in a strand exchange reaction. The DNA-binding mutant fragment (146AAAA) did not support strand exchange on its own and in the presence of RAD51 (*Figure 5—figure supplement 1*).

Full length PALB2 promotes D-loop formation by RAD51 (*Buisson et al., 2010*; *Dray et al., 2010*). However, we found that N-DBD did not support D-loop formation with ssDNA oligonucleotide and supercoiled dsDNA (*Figure 4—figure supplement 4*).

### RPA inhibits PALB2-mediated strand exchange

RPA inhibits formation of the presynaptic RAD51 filament on ssDNA and recombination mediator proteins RAD52 and BRCA2/DSS1 overcome this inhibitory effect (*Benson et al., 1998*; *Jensen et al., 2010*; *Liu et al., 2010*; *New et al., 1998*; *Plate et al., 2008*; *Shinohara and Ogawa, 1998*; *Sugiyama and Kowalczykowski, 2002*; *Sung, 1997*; *Thorslund et al., 2010*; *Zhao et al.,*

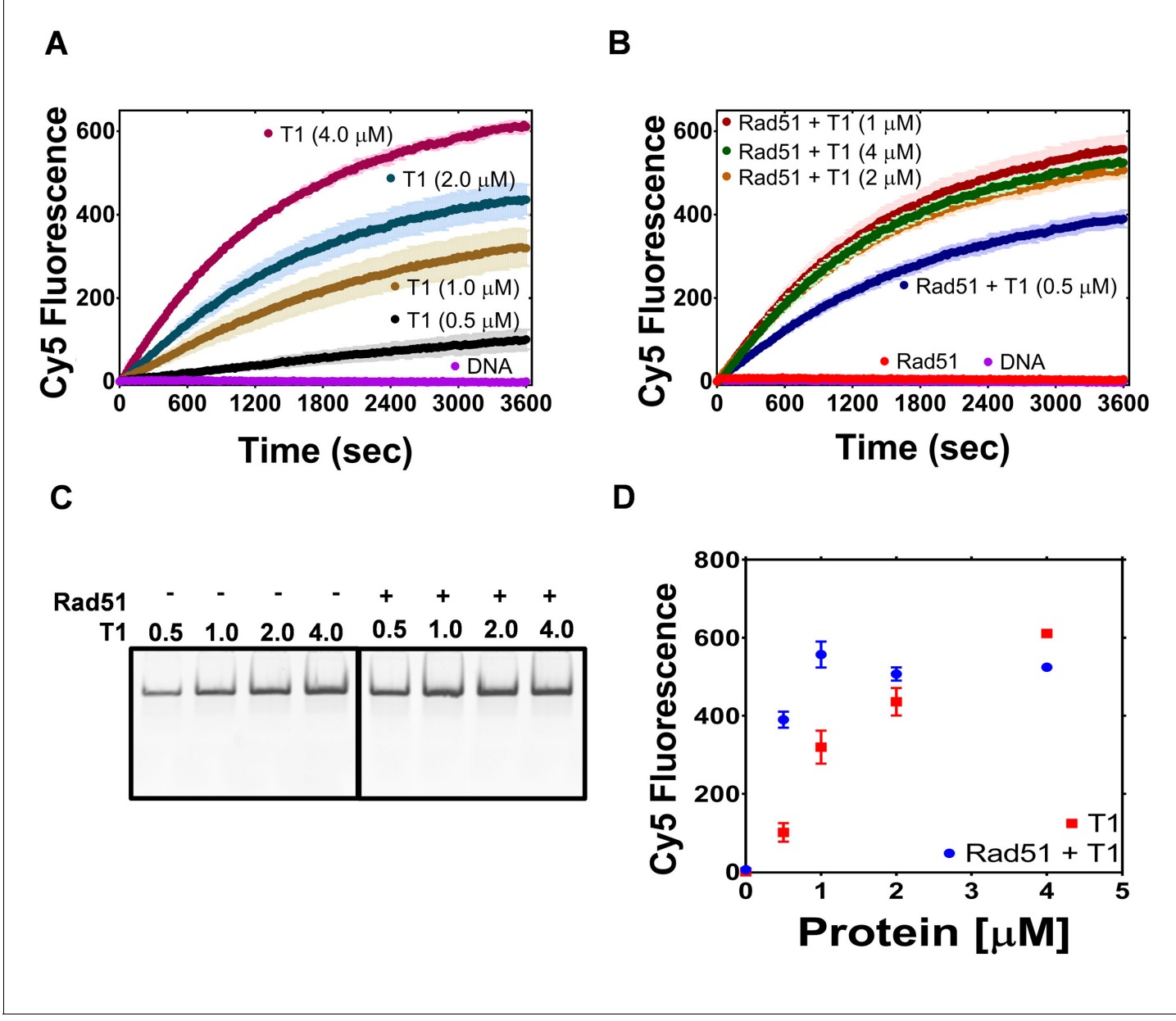

**Figure 5.** The PALB2 N-DBD stimulates RAD51 strand exchange. (**A**) Dependence of the strand exchange activity on N-DBD concentration: 0.5 µM (black), 1 µM (brown), 2 µM (navy) and 4 µM (dark red). (**B**) A reaction similar to (**A**) in the presence of RAD51 (2 µM, red: without N-DBD). Each point is an average of three measurements in (**A**) and (**B**). (**C**) Deproteinated strand exchange activity products from (**A**) and (**B**) separated on native PAGE gel. (**D**) End-point values of the strand exchange reactions shown in (**A**) and (**B**) plotted against N-DBD concentration. Buffer and DNA concentration are identical to those in *Figure 4*.

DOI: https://doi.org/10.7554/eLife.44063.020

The following figure supplement is available for figure 5:

**Figure supplement 1.** PALB2 DNA binding site mutants do not support strand exchange.

DOI: https://doi.org/10.7554/eLife.44063.021

*2015*). In both cases, the mediation reaction depends on protein interaction with RPA. PALB2 does not interact with RPA. Correspondingly, we found that RPA has a strong inhibitory effect on the strand exchange reaction by PALB2 DBD alone and in the presence of RAD51 (*Figure 6*).

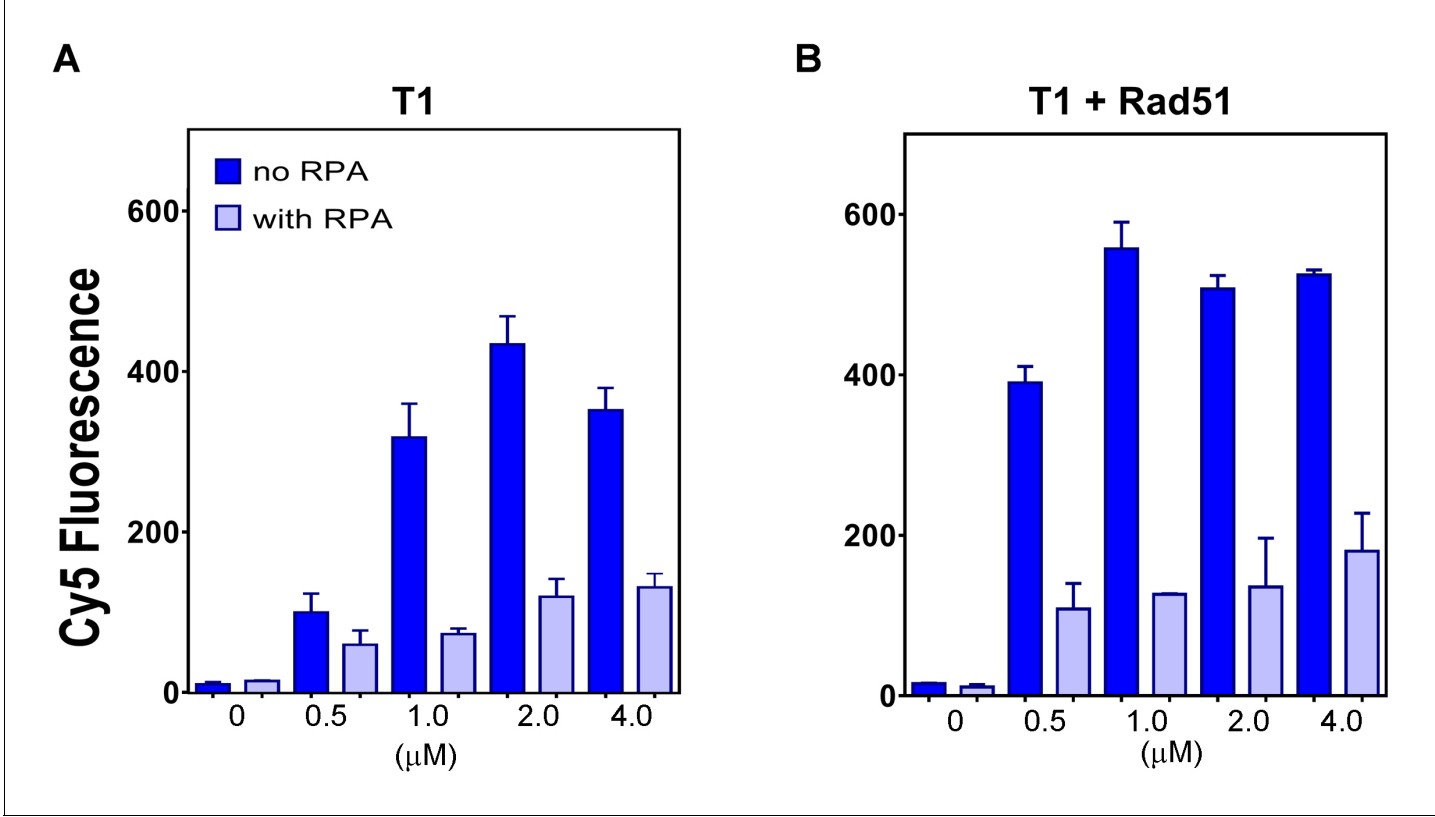

**Figure 6.** The plot of end points of strand exchange reactions with and without 0.5 μM RPA by (**A**) N-DBD and (**B**) N-DBD in the presence of 2 μM RAD51. Reactions are performed identically to those in *Figure 5*.

DOI: https://doi.org/10.7554/eLife.44063.022

## PALB2 stimulates an inverse strand exchange and can use an RNA substrate

The strand exchange capability of PALB2 N-DBD resembles those of RAD52. RecA and Rad52 support an inverse strand exchange as well as R-loop formation (*Kasahara et al., 2000*; *Mazina et al., 2017*; *Zaitsev and Kowalczykowski, 2000*). Functional significance of this was demonstrated for an RNA-templated DSB repair (*Mazina et al., 2017*). Therefore, we tested the PALB2 N-DBD for similar activities. The PALB2 N-DBD supported both forward and inverse strand exchange with similar efficiencies (*Figure 7B,E*). Furthermore, PALB2 supported both reactions with a ssRNA substrate (*Figure 7C,F*). RAD52 was shown to have different efficiencies of forward and inverse reactions with relatively low forward and a more efficient inverse reactions (*Mazina et al., 2017*). We did not observe this difference with PALB2. The inverse strand exchange was slower than in case of RAD52 and comparable to that of RAD51 under optimal conditions. However, the substrates used in current work and in RAD52 studies are different.

## Mechanism of the PALB2 stimulated strand exchange

To rule out a potential effect of DNA melting by PALB2, which may lead to nonspecific reannealing of a separated strands with complementary ssDNA in solution, the N-DBD was incubated with dsDNA without ssDNA (*Figure 8A*). The N-DBD does not melt dsDNA as there was no change in fluorescence of Cy5/Iowa-ds35 upon incubation with the protein in the absence of ssDNA, addition of which triggers the reaction. Moreover, N-DBD stimulates annealing of complementary ssDNA (*Figure 8B*). Therefore, the observed strand exchange is not a consequence of a nonspecific dsDNA melting by the protein.

Both RecA and RAD52 proteins, which support strand exchange, simultaneously interact with ds- and ssDNA through distinct binding sites located next to each other (*Arai et al., 2011*; *Chen et al., 2008*; *Honda et al., 2011*; *Kagawa et al., 2002*; *Mazin and Kowalczykowski, 1998*; *Seong et al.,*

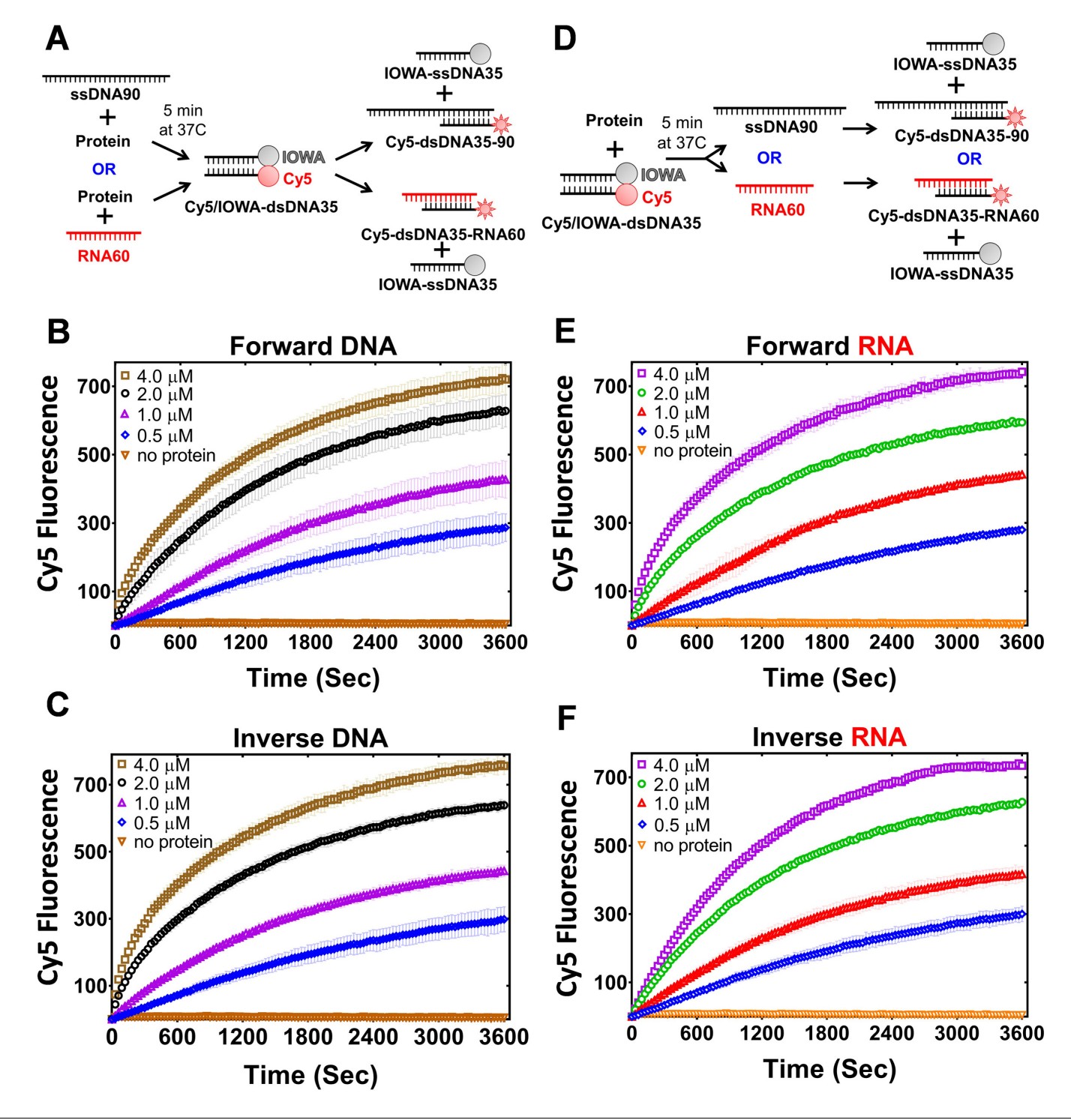

**Figure 7.** PALB2 promotes forward and inverse strand exchange with ssDNA and RNA substrates. Schematic representation of forward (**A**) and inverse (**D**) reactions. Cy5 fluorescence change for the forward reaction with ssDNA is shown in (**B**) and with RNA in (**E**) at different concentrations of T1 fragment ranging from 4 µM (brown in B, magenta in E) to 0.5 µM (blue) and without protein in orange. Similar time courses of inverse reactions for ssDNA are shown in (**C**) and for RNA in (**F**). Buffer and DNA concentration are identical to those in *Figure 4*. Each point is an average of three measurements.

DOI: https://doi.org/10.7554/eLife.44063.023

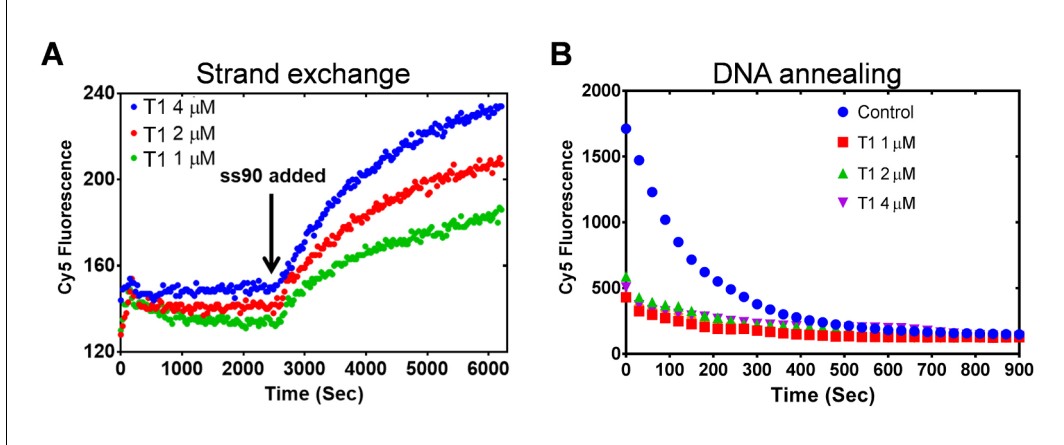

**Figure 8.** PALB2 does not unwind dsDNA and anneals complementary DNA strands. (**A**) Strand exchange reaction where Cy5/Iowa ds35 DNA (100 nM) was first incubated with three different concentrations of N-DBD for 30′. The complementary ss90 DNA (100 nM) was added at 30′ to initiate the strand exchange. (**B**) Annealing of Cy5- and Iowa-labelled complementary ss35 strands (100 nM) in the presence of different concentrations of PALB2 N-DBD. Buffers are identical to those in *Figure 4*.

DOI: https://doi.org/10.7554/eLife.44063.024

The following figure supplements are available for figure 8:

**Figure supplement 1.** RecO and RecOR do not support strand exchange without RecA.

DOI: https://doi.org/10.7554/eLife.44063.025

**Figure supplement 2.** Oligomerization and DNA-binding stoichiometry of PALB2 N-DBD.

DOI: https://doi.org/10.7554/eLife.44063.026

---

*2008*). PALB2 also interacts with both ss- and dsDNA (*Figure 1* and *Buisson et al., 2010*; *Dray et al., 2010*), although the structure and the molecular details of PALB2 interaction with DNA remain unknown. To verify if other proteins characterized by comparable affinities to both ss- and dsDNA also support strand exchange, we tested prokaryotic RMPs, RecO and RecR. *E. coli* RecO alone stimulates strand annealing (*Kantake et al., 2002*; *Luisi-DeLuca and Kolodner, 1994*) and, in complex with RecR, stimulates RecA-mediated strand exchange with ssDNA bound to SSB (*Ryzhikov et al., 2014*; *Umezu et al., 1993*; *Umezu and Kolodner, 1994*). Both RecO and RecOR interact with ss- and dsDNA (*Ryzhikov et al., 2014*). However, neither RecO nor RecOR complex promote strand exchange in the absence of RecA (*Figure 8—figure supplement 1*). Therefore, the simple ability of a protein to interact with ss- and dsDNA is not enough to promote strand exchange and even RMPs, which stimulate the reaction by RecA recombinase, do not support it in the absence of recombinase.

RecA/RAD51 and RAD52 proteins form oligomeric structures, such as recombinase-DNA filament (*Chen et al., 2008*; *Egelman and Stasiak, 1986*; *Yang et al., 2001*) or Rad52 ring structure (*Shinohara et al., 1998*; *Singleton et al., 2002*). Both N-DBD (*Figure 8—figure supplement 2*) and the full length PALB2 form oligomeric structures (*Buisson and Masson, 2012*; *Sy et al., 2009c*). The oligomerization is partially mediated by the N-terminal coiled-coil motif (*Sy et al., 2009c*). Thirty N-terminal amino acids form an antiparallel coiled-coil dimer (*Song et al., 2018*). However, gel filtration experiments suggest a tetrameric form (*Figure 8—figure supplement 2A*). The titration of double-labelled Cy5/Cy3 ssDNA performed similar to the experiment reported in *Grimme et al. (2010)* revealed maximum FRET value at 1:4 or 1:5 ratio for both 70 nt and 40 nt long ssDNA (*Figure 8—figure supplement 2B*). The titration of ss49 by the N-DBD also suggests a stoichiometry of four or five N-DBD monomers per ss49 (*Figure 8—figure supplement 2C,D*). Importantly, the elution volume of the full length PALB2 in previous report also corresponds to a significantly higher molecular weight than that of a dimer, and the molecular weight of PALB2 with truncated coiled-coil domain higher than that of a monomer. Therefore, we expect to identify a secondary oligomerization site in N-DBD which should support a tetrameric structure.

## Discussion

We identified major DNA-binding residues of PALB2 and demonstrated their critical role for HDR in cells. PALB2 is described as a scaffold protein linking BRCA1 with BRCA2 during HDR and interacting with many other chromatin proteins. However, the truncation variant with the preserved BRCA1 and BRCA2 binding motifs but without the middle portion of the protein, which contains DBDs, does not support BRCA2 and RAD51 recruitment to DSBs (*Sy et al., 2009b*). Recent studies suggest an alternative BRCA1-independent recruitment of PALB2 to DSB through direct interaction with RNF168 (*Luijsterburg et al., 2017*; *Zong et al., 2019*). However, this interaction is mediated by the C-terminal WD40 domain, which is preserved in the described above truncation mutant. A critical role of PALB2 DNA binding was also suggested by studies of the BRCA2 (*Siaud et al., 2011*), where the 'miniBRCA2' construct, including only DBDs with two BRC repeats, was 3–4 times less efficient than 'midiBRCA2' which includes PALB2 interaction motif. Moreover, interaction with PALB2 alleviates the requirement of BRCA2 DNA binding, including a deletion of the entire BRCA2 DBD. Here, we demonstrate that mutation of only four DNA-binding residues of PALB2 reduces both RAD51 foci and overall HDR efficiency by 50% in the presence of endogenous BRCA1 and BRCA2. Even at 8 hr post-IR siRNA PALB2 cells complemented with PALB2-146AAAA still display a defect in HR. We suggest that the mutant display a failure to repair a subset of DSBs due to its defective DNA binding. Therefore, PALB2 interaction with DNA is critical for recruitment of the BRCA2 and RAD51 to DSB sites and efficient DNA repair in cells.

Secondly, we demonstrate that the PALB2 N-DBD stimulates RAD51-mediated strand exchange in vitro. N-DBD encompasses the RAD51 binding site, therefore, it can recruit RAD51 to ssDNA to initiate nucleation of nucleoprotein filament. DNA-binding of N-DBD plays a critical role in this process, as the DNA-binding mutant (146AAAA) retains interaction with RAD51 (*Figure 3—figure supplement 3*) but is unable to stimulate RAD51 binding to ssDNA (*Figure 5—figure supplement 1*). Alternatively, the PALB2 N-DBD can support a DNA conformation favourable for RAD51 binding to ssDNA or/and nucleation of RAD51 filament, similarly to the prokaryotic RecOR proteins (*Bell et al., 2012*; *Ryzhikov et al., 2014*; *Sakai and Cox, 2009*).

The most unexpected finding is the ability of PALB2 to stimulate strand exchange between homologous ss- and dsDNA fragments in the absence of recombinase. This process is protein-specific and is not a consequence of simple DNA melting and reannealing of separated strands in solution, since the PALB2 N-DBD does not unwind the DNA helix and promotes DNA annealing. PALB2-mediated strand exchange differs from that of supported by RAD51, as it does not require ATP binding and hydrolysis and it is less efficient and not unidirectional. This reaction resembles that of the RAD52 (*Mazina et al., 2017*). In the cited manuscript, RAD52 promotes inverse reaction much more efficiently than forward and seemingly faster than that by PALB2 in our hands. However, the substrates used in RAD52 studies differ from those used in the current work by the presence of the ssDNA tail in dsDNA. The reaction with a no-tail substrate appears to be less efficient than the one supported by PALB2. Another distinction from RAD52 is the initial reaction rate, which was much faster in case of inverse reaction even with no-tail substrate for RAD52. PALB2-mediated reactions are slower with no difference between forward and inverse reactions.

Proteins supporting strand exchange, such as RecA and RAD52, share several common features. They interact with both ss- and dsDNA through distinct sites located next to each other, they form oligomeric structures, such as recombinase-DNA filament or Rad52 ring structure, and they distort the dsDNA helix to initiate strand exchange with the bound complementary ssDNA. RecA stretches dsDNA (*Chen et al., 2008*; *Leger et al., 1998*), while Rad52 bends the DNA helix bound to the toroidal oligomeric ring (*Brouwer et al., 2017*). The PALB2 N-DBD interacts with both ss- and dsDNA. Previously, we demonstrated that PALB2 immobilized on ssDNA beads efficiently pulls down non-homologous dsDNA (*Buisson et al., 2010*). It is unclear whether ss- and dsDNA substrates are bound to the same sites of different subunits of a PALB2 oligomer or two different DNA binding sites on the same monomer. The presence of at least two other minor DNA-binding sites in the N-DBD suggests such a possibility. Higher affinity towards longer ssDNA (*Figure 1*) and the FRET experiment (*Figure 8—figure supplement 1B*) support a model of wrapping long flexible ssDNA around an oligomer. In contrast, interaction with dsDNA is less length dependent (*Figure 1*). We can speculate that binding of dsDNA to more than one monomer in PALB2 oligomer can trigger DNA helix distortion. Thus, PALB2 shares several specific structural and DNA-interaction features

with both RecA/RAD51 and Rad52 proteins and supports a protein-specific strand exchange reaction. At the same time, PALB2 does not support D-loop formation with supercoiled dsDNA opposite to a full length PALB2. This may be a consequence of the inability to efficiently bind and distort supercoiled dsDNA plasmid through the secondary DNA-binding site.

It is important to note one distinct feature of the PALB2 N-DBD: the secondary structure prediction (*Figure 2—figure supplement 1*) suggests a different folding of the N-DBD fragment than that of RecA-like domains or a Rad52. The latter proteins are formed by α/β sandwich folds, while PALB2 N-DBD folding is predicted to be composed of only α-helices, which may form helical bundle-like structure similar to that of Hop2-Mnd1 (*Kang et al., 2015*). Therefore, PALB2 N-DBD represents a novel structural fold that supports strand exchange.

While the functional significance of this property remains to be investigated, it further supports the involvement of PALB2 in specific DNA transactions during HDR, similar to the involvement of BRCA1/BARD1 in D-loop formation (*Zhao et al., 2017*). It was shown that both PALB2 and BRCA2 stimulate Polη DNA synthesis within a D-loop substrate in vitro through the recruitment of the polymerase to the invading strand in the D-loop (*Buisson et al., 2014*). Interestingly, DNA synthesis was more efficient in the presence of PALB2 than BRCA2, while both proteins were shown be equally efficient in recruiting polymerase to DSB sites. PALB2 strand exchange may contribute to other steps of HDR such as second-end capture (*Mazloum and Holloman, 2009*; *McIlwraith and West, 2008*; *Nimonkar et al., 2009*). Interestingly, PALB2 (FANCN) and BRCA2 (FANCD) are involved in replication-dependant removal of interstrand DNA crosslinks associated with Fanconi anemia (*Howlett et al., 2002*; *Moldovan and D'Andrea, 2009*; *Xia et al., 2007*). The strand exchange function of PALB2 may also be important for alternative DNA repair pathways. Indeed, PALB2 supports strand exchange not only with ssDNA, but with ssRNA substrates, and can be involved in transcription-initiated DNA repair. This hypothesis seems particularly attractive in light of the PALB2 interaction with MRG15 and its localization at the sites of actively transcribed genes *Bleuyard et al. (2017)*.

# Materials and methods

### Key resources table

| Reagent type (species) or resource | Designation | Source or reference | Identifiers | Additional information |
|---|---|---|---|---|
| Cell line (Human) | HeLa | ATCC | CCL-2 | |
| Cell line (Human) | U2OS | ATCC | HTB-96 | |
| Cell line (Human) | HEK293T | ATCC | CRL-11268 | |
| Sequence-base reagent | siRNA sequences | This paper | | *Supplementary file 1E* |
| Sequence-base reagent | oligonucleotide JYM1413 | This paper | | *Supplementary file 1C* |
| Chemical compound, drug | Lipofectamine RNAiMAX | Invitrogen | 13778–150 | |
| Chemical compound, drug | Thymidine | Sigma | T1895-1G | |
| Chemical compound, drug | Lipofectamine 2000 | Invitrogen | 11668019 | |
| Chemical compound, drug | Proteinase K | Sigma | P2308-100MG | |
| Chemical compound, drug | [gamma-32P] ATP | Perkin Elmer | NEG502A250UC | |
| Recombinant DNA reagent | YFP-CTL or YFP (plasmid) | *Pauty et al., 2017* | | |
| Recombinant DNA reagent | YFP-PALB2 (plasmid) | *Pauty et al., 2017* | | |

*Continued on next page*

*Continued*

| Reagent type (species) or resource | Designation | Source or reference | Identifiers | Additional information |
|---|---|---|---|---|
| Recombinant DNA reagent | YFP-PALB2 146AAAA (plasmid) | This paper | | *Supplementary file 1B* |
| Recombinant DNA reagent | pCR2.1-mClover-LMNAdonor (plasmid) | *Pinder et al., 2015* | | |
| Recombinant DNA reagent | pX330-LMNAgRNA (plasmid) | *Pinder et al., 2015* | | |
| Recombinant DNA reagent | iRFP670 (plasmid) | *Pinder et al., 2015* | | |
| Recombinant DNA reagent | FLAG vector (plasmid) | *Pauty et al., 2017* | | |
| Recombinant DNA reagent | FLAG-PALB2 (plasmid) | *Pauty et al., 2017* | | |
| Recombinant DNA reagent | FLAG-PALB2 146AAAA (plasmid) | This paper | | *Supplementary file 1B* |
| Recombinant DNA reagent | pPB4.3 | PMID:15899844 | | |
| Other | DAPI stain | Invitrogen | D1306 | (1 µg/mL) |
| Commercial assay, kit | SE Cell Line 4D-Nucleofector X Kit | VWR | CA10064-148 | |
| Antibody | anti-RAD51 (mouse monoclonal) | Novus Biologicals | NB100-148 | (1:1000) |
| Antibody | anti-GFP (mouse monolconal) | Roche | 11814460001 | (1:1000) |
| Antibody | RAD51 (rabbit polyclonall) | B-Bridge International | 70–001 | (1:7000) |
| Antibody | cyclin A (mouse monoclonal) | BD Biosciences | 611268 | (1:400) |
| Antibody | Alexa Fluor 568 goat anti-rabbit | Molecular Probe | A11011 | (1:10000) |
| Antibody | Alexa Fluor 647 got anti-mouse | Molecular Probe | A21235 | (1:10000) |
| Software, algorithm | Prism | GraphPad | Ver-6 | |
| Software, algorithm | Volocity | Quorum Technologies | v6.0.1 | |
| Sequence-base reagent | cloning primers, DNA-binding substrates, FRET substrates | IDT, Technology | | *Supplementary file 1B,C,D* |
| Strain, Strain background (*E. Coli*) | BL21-star cells | ThermoFisher Scientific | C601003 | |
| Strain, Strain background (*E. Coli*) | OmniMAX cells | ThermoFisher Scientific | C8540-03 | |
| Recombinant DNA reagent | pSMT3-PALB2-T1 | This paper | | |
| Recombinant DNA reagent | pSMT3-PALB2-T3 | This paper | | |
| Recombinant DNA reagent | pSMT3-PALB2-573 | This paper | | |
| Other | His60 Ni Superflow | Clonetech | 635660 | |
| Other | Heparin 5 ml HiTrap column | GE Heathcare Lifesciences | 17040701 | |
| Other | Superdex-200 10/300 GL column | GE Heathcare Lifesciences | 17517501 | |

*Continued on next page*

*Continued*

| Reagent type (species) or resource | Designation | Source or reference | Identifiers | Additional information |
|---|---|---|---|---|
| Recombinant DNA reagent | pET11-Rad51 | Dr. A Mazin Lab; PMID:11751636 | | |
| Recombinant DNA reagent | pSMT3-RAD51 | This paper | | |
| Recombinant DNA reagent | p11d-tRPA | AddGene, *Henricksen et al., 1994* | 102613 | |
| Other | Affi-Gel blue affinity gel | BioRad | 1537301 | |
| Instrument | Synergy four plate reader | BioTek | | |

## Protein purification

### PALB2 truncations

PALB2 N-terminal fragments PALB2-T1 (1–200 aa) and PALB2-573 (1–573 aa) were cloned into pET28b + based pSMT3 vector (provided by Dr. R.A. Kovall, University of Cincinnati) containing the N-terminal 6xHis-SUMO tag using *Sal*I and *Nde*I cloning sites. pSMT3-PALB2 T1, pSMT3-PALB2 573 were transformed into BL21* cells. Cell cultures were grown in LB to OD600 = 0.7 and protein expression was induced by adding 0.2 mM IPTG and carried out at 16°C overnight. Cells were lysed with lysozyme (0.25 mg/mL at RT for 30 min) in lysis buffer (25 mM HEPES pH 8.0, 1 M NaCl, 10% glycerol, 0.3% Brij35, 1 mM TCEP, 2 mM CHAPS and 1 mM PMSF), followed by three rounds of sonication (50% output and 50% pulsar settings for 4 min). Cell debris were removed by centrifugation at 30,600 x g for 45 min. Supernatant was loaded on a NiNTA column (5 ml) equilibrated with binding buffer (25 mM HEPES pH 8.0, 1 M NaCl, 10% glycerol, 1 mM TCEP, 2 mM CHAPS and 10 mM imidazole). NiNTA beads were washed with binding buffer and the protein was eluted with binding buffer supplemented with the same buffer adjusted to 250 mM imidazole. The SUMO tag was cleaved with Ulp1 protease while dialyzing against buffer without imidazole (25 mM HEPES pH 8.0, 1 M NaCl, 10% glycerol, 1 mM TCEP and 2 mM CHAPS) overnight and the protein was purified with a second NiNTA column. The protein was diluted 10X by binding buffer without NaCl to the final NaCl concentration of 100 mM, loaded to a Hi-Trap heparin affinity column (5 ml, GE health sciences) and eluted with a gradient of NaCl (100 mM to 1000 mM). Protein eluted from the heparin column at ~500 mM NaCl concentration. Protein fractions were dialysed against storage buffer (25 mM HEPES pH 8.0, 300 mM NaCl, 40% glycerol, 1 mM TCEP and 2 mM CHAPS) overnight, aliquoted and stored at −80°C.

PALB2 T3 fragment was purified as described in *Buisson et al. (2010)*.

### RAD51 purification

We used two expression constructs and purification protocols. (1) Human RAD51 protein was purified from the pET11-Rad51 vector (gift from Dr A. Mazin) according to the published protocol (*Sigurdsson et al., 2001*). The protein was induced at 37°C for 3 hr by supplementing LB media with 0.5 mM IPTG. Cells were suspended in 25 mM Tris-HCl pH8.0, 1 M urea, 1 M NaCl, 5 mM DTT, 0.3% Brij35% and 10% glycerol. Cells were lysed with lysozyme (0.25 mg/mL at RT for 30 min) followed by three rounds of sonication (50% output and 50% pulsar settings for 1 min). 24 mg/ml ammonium sulfate was gradually added to the supernatant and equilibrated overnight at 4°C. Precipitates were centrifugation at 30,600 x g for 45 min. Pellets were solubilized in 30 ml of binding buffer (25 mM Tris-HCl pH8.0, 1 M NaCl, 5 mM DTT, 10% glycerol and 20 mM imidazole). Insoluble particles were removed by centrifugation at 30,600 x g for 40 min. The protein was bound to Ni NTA beads, extensively washed with binding buffer and eluted in binding buffer supplemented with 250 mM imidazole.

(2) Alternatively, human RAD51 gene was cloned into pSMT3 vector using *Sal*I and *Nde*I cloning sites. pSMT3-Rad51 protein expression was carried out at 16°C overnight by addition of 0.2 mM IPTG. SUMO tagged Rad51 protein was purified according to the steps described for the PALB2 fragments. Purified Rad51 protein was dialysed against storage buffer (25 mM HEPES pH 8.0, 300 mM NaCl, 40% glycerol, 1 mM TCEP and 2 mM CHAPS) overnight, aliquoted and stored in −80°C

until further use. Proteins from both preparations had comparable properties. Data are shown for experiments performed with the second construct, except for *Figure 5—figure supplement 1*.

*E. coli* RecA was purified exactly as described in *Gupta et al. (2013)*. *E. coli* RecO and RecR proteins were purified as described (*Ryzhikov and Korolev, 2012*; *Ryzhikov et al., 2011*). Protein concentration was determined by Bradford reagent (Thermo scientific).

Trimeric RPA complex was expressed using p11-tRPA vector (Addgene, plasmid #102613) and purified accordingly to the published protocol (*Binz et al., 2006*; *Henricksen et al., 1994*). A fresh transformant of p11d-tRPA was grown to 0.5 OD at 37°C in a litre of LB media. Protein expression was induced by IPTG (0.2 mM) at 16°C overnight. Cells were harvested and suspended in lysis buffer (25 mM HEPES pH 8.0, 0.25 mM EDTA, 1 mM PMSF, 1 mM TCEP and 0.01% NP-40). Cells were lysed by incubation with lysozyme (0.25 mg/mL at 25°C for 30 min) and followed by sonication (50% output and 50% pulsar settings for 4 min). Cell debris were removed, and supernatant was loaded on an Affi-Gel Blue column (5 ml, Biorad) equilibrated with lysis buffer. The column was washed sequentially with five column volumes of lysis buffer containing 50 mM KCl, 0.8 M KCl, 0.5 M NaSCN and 1.5 M NaSCN. RPA elutes in the 1.5 M NaSCN wash. This fraction was diluted ten times to a final concentration of NaSCN to 150 mM and applied to heparin column. The column was washed with lysis buffer followed by elution with a linear gradient of NaCl (0.1–1M). The peak fractions were pooled, concentrated, flash frozen in liquid nitrogen and stored at −80°C until further use.

## Site-directed mutagenesis

Target amino acids were mutated by site directed mutagenesis using Stratagene QuikChange protocol. Single, double, triple and four residues mutants were generated by single stranded synthesis (*Supplementary file 1B*). PCR samples were subjected to *Dpn*I digestion at 37°C for 6 hr and annealed gradually by reducing temperature from 95°C to 37°C for an hour with a 1°C drop per minute. *Dpn*I treated PCR samples were transformed into chemically competent OmniMAX cells (ThermoFischer). Mutations were confirmed by sequencing and plasmids were transformed into BL21(DE) cells. Mutant proteins were expressed and purified exactly as described for wild type fragments.

## DNA binding assay

Fluorescence anisotropy experiments were carried out at room temperature with 5 nM fluorescein (6FAM)-labelled DNA substrates (*Supplementary file 1C*) using a Synergy four plate reader (BioTek). Titration with protein was performed by serially diluting protein in 40 µL of assay buffer (20 mM Tris-acetate pH 7.0, 100 mM NaCl, 5% glycerol, 1 mM TCEP and 10% DMSO) from 5000 nM to 0.3 nM and incubating with DNA substrate for 15 min at RT. Fluorescence anisotropy was measured by excitation at 485/20 nm and by monitoring emission 528/20 nm at room temperature using Gen5.0 (Bio-Tek) software. An equilibrium dissociation constant was calculated with Prism software using one to one binding scheme, $[P] + [D] = [DP]$, where D is DNA and P is PALB2. Anisotropy data were fitted by a non-linear regression analysis of Prism software using standard four-parameter logistic equation to identify $K_d$

$$y = y_{min} + \left( \frac{y_{max} - y_{min}}{1 + 10^{(logEC50 - X) \times n}} \right)$$

Where, $y_{min}$ and $y_{max}$ is the minimum and maximum anisotropy values, X represents the log concentration of protein, $n$ represents Hillslope, and EC50 is equal to $K_D$. $R^2$ is determined by the prism software by computing the sum of the squares of the distances of the points from the best-fit curve determined by nonlinear regression model.

## DNA annealing assay

DNA annealing assays were performed with Cy5-labelled ss35 and compl IOWA-labelled ss35 (100 nM, *Supplementary file 1D*). The protein at 1, 2 and 4 µM concentrations was mixed and incubated with Cy5-labelled ssDNA for 5 min at 37°C in 40 µL reaction buffer (40 mM HEPES pH 7.5, 20 mM NaCl and 1 mM TCEP). Reactions were initiated by addition of compl IOWA-labelled ss35 (100 nM) in 40 µL of reaction buffer. Decrease in Cy5 fluorescence was monitored by measuring fluorescence at 680 nm by excitation at 635 nm on a Synergy four plate reader (BioTek).

## Strand exchange fluorescent assay

DNA strand exchange assays (80 µl) were performed with 35 bp dsDNA obtained by annealing of 5'-Cy5- and 3'-IOWA-labelled compl strands (*Supplementary file 1D*) and a 90mer ssDNA (ss90) with a region homologous to the plus strand. Alternatively, FAM/Dabsyl 49 bp DNA was used. For the forward reaction, ss90 (100 nM) was incubated with 2 µM (or as mentioned in the figure legends) protein for 10 min in 40 µL reaction buffer (40 mM HEPES pH 7.5, 20 mM NaCl, 5 mM MgCl$_2$, 1 mM TCEP and 0.02% Tween 20) at 37°C. Strand exchange was initiated by addition of 100 nM Cy5/IOWA-dsDNA35 (or FAM/Dabsyl-dsDNA49), the plate was immediately placed in plate reader and the intensity of Cy5 (or FAM) fluorescence was measured at 30 s intervals for 1 hr with excitation at 635 nm and emission at 680 nm. For reactions with RecA and Rad51, an ATP regeneration system (2 mM ATP, 3 mM phosphoenol pyruvate and 30 U of pyruvate kinase) was used (Sigma-Aldrich, USA). Experiments without RecA and RAD51 were performed without ATP. For the inverse reaction, protein was incubated with Cy5/IOWA-dsDNA35 (or FAM/Dabsyl-dsDNA49) substrate and reaction was initiated by addition of ss90. The strand exchange assay with ssRNA substrate was performed as described above using a 60 ribonucleotide RNA (*Supplementary file 1D*) compl to that of 35 bp DNA. Alternatively, Cy3- and Cy5-labelled DNA oligonucleotides were used to prepare dsDNA substrate and the products were analysed by EMSA PAGE (below).

## EMSA PAGE

Fluorescent-labelled DNA products of strand exchange reactions were also analysed on EMSA PAGE. After fluorescence measurement on plate reader, the final reaction mix (80 µl) products were deproteinated by incubation with proteinase K (0.5 mg/ml) with 0.5 mM EDTA and 1% w/v SDS for 20 min at 37°C and the DNA fragments were separated on 10% PAGE gel in TBE buffer. The gel was imaged using a Typhoon 9400 image scanner (GE) and analysed with ImageJ software.

## FRET assay

FRET assay was performed in 96 well plate format. 100 nM of dual labelled dT$_{70}$ or ss40 (Cy5 at 5' end and Cy3 at 3' end) was dispensed into 80 µL assay buffer identical to the buffer in the strand exchange assay (*Supplementary file 1D*). Excitation was at 540/25 nm bandpass. Emission for both Cy3 at 590/35 nm bandpass and Cy5 at 680/30 nm were measured. PALB2 was added with final concentration between 4000 and 15.6 nM and incubated for 10 min. FRET efficiency was calculated by using the formula $FRET = \frac{1.51 \times Icy5}{(1.51 \times Icy5 + 0.669 \times Icy3)}$, where 1.51 and 0.669 are correction factors for fraction of Cy5 and Cy3 intensities in Cy3 and Cy5 channels respectively. Protein concentrations were as described in the figure legends.

## D-loop assay

D-loop buffer (25 mM Tris-acetate pH 7.5, 100 µg/mL bovine serum albumin, 2 mM CaCl$_2$, 2 mM ATP, 1 mM DTT), containing 1 µM of radiolabelled oligonucleotide JYM1413 (homologous sequence to the plasmid pPB4.3, *Supplementary file 1C*) was incubated for 5 min at 37°C with the indicated concentration of RAD51 or PALB2-T1 fragment. CsCl- purified pPB4.3 replicative form I DNA (300 µM) was added and the reaction and incubated for 5 min. Finally, the reaction was stopped with the addition of 0,6% SDS, 20 mM Tris-HCl pH 7.5, 20 mM MgCl$_2$ and 2 mg/mL proteinase K following by incubation for 30 min at 37°C. Labelled DNA products were analysed by electrophoresis through a 0.8% TAE1X/agarose gel, ran at 65V for 1 hr, dried onto DE81 filter paper at 85°C, and visualized by autoradiography.

## RAD51 foci assay

HeLa cells were seeded on glass coverslips in 6-well plates at 225 000 cells per well. Knockdown of PALB2 was performed 18 hr later with 50 nM PALB2 siRNA (*Supplementary file 1E*) using Lipofectamine RNAiMAX (Invitrogen). After 5 hr, cells were subjected to double thymidine block. Briefly, cells were treated with 2 mM thymidine for 18 hr and released after changing the media.

After a release of 9 hr, PALB2 silenced cells were complemented using transfection with the indicated YFP constructs using Lipofectamine 2000. Cells were then treated with 2 mM thymidine for 17 hr and protected from light from this point on. After 2 hr of release from the second block, cells were X-irradiated with 2 Gy and processed for immunofluorescence 1 hr, 2 hr, 4 hr, 6 hr and 8 hr

post-irradiation. All immunofluorescence dilutions were prepared in PBS and incubations performed at room temperature with intervening washes in PBS. Cell fixation was carried out by incubation with 4% paraformaldehyde for 10 min followed by 100% ice-cold methanol for 5 min at −20°C. Cells were then permeabilized in 0.2% Triton X-100 for 5 min and quenched using 0.1% sodium borohydride for 5 min. After blocking for 1 hr in a solution containing 10% goat serum and 1% BSA, cells were incubated for 1 hr with primary antibodies to RAD51 (B-bridge International, #70–001) and to cyclin A (BD Biosciences, # 611268) diluted in 1% BSA. Secondary antibodies, Alexa Fluor 568 goat anti-rabbit (Invitrogen, #A-11011) and Alexa Fluor 647 goat anti-mouse (Invitrogen, #A-21235), were used in PBS containing 1% BSA for 1 hr. Nuclei were stained for 10 min with 1 μg/mL 4, 6-diami-dino-2-phenylindole (DAPI) prior to mounting onto slides with 90% glycerol containing 1 mg/ml par-aphenylenediamine anti-fade reagent. Z-stack images were acquired at 63X magnification on a Leica DM6000 microscope, then deconvoluted and analysed for RAD51 foci formation with Volocity soft-ware v6.0.1 (Perkin-Elmer Improvision). The number of RAD51 foci per cyclin A-positive cells (n = 300), among the transfected population, was manually scored and reported in a scatter dot plot representing the SEM. An Anova test (Kruskal-Wallis-test for multiple comparison) was performed followed by a non-parametric Mann-Whitney test.

## CRISPR-Cas9/mClover-LMNA1 mediated HR assay (*Pinder et al., 2015*)

U2OS cells were seeded in 6-well plates. Knockdown of PALB2 (*Buisson et al., 2017*) was performed 6–8 hr later using Lipofectamine RNAiMAX (Invitrogen). Twenty-four hours post-transfection, $1.5-2 \times 10^6$ cells were pelleted for each condition and resuspended in 100 μL complete nucleofector solution (SE Cell Line 4D-Nucleofector X Kit, Lonza) to which 1 μg of pCR2.1-mClover-LMNAdonor, 1 μg pX330-LMNAgRNA, 0.1 μg of iRFP670 and 1 μg of pcDNA3 empty vector or the Flag-PALB2 constructs, and 20 nM of each siRNA were added. Once transferred to a 100 μl Lonza certified cuvette, cells were transfected using the 4D-Nucleofector X-unit, program CM-104 and transferred to a 10 cm dish. After 48 hr, cells were trypsinized and plated onto glass coverslips. Expression of the mClover was assayed the next day by fluorescence microscopy (63X), that is 72 hr post-nucleofection. Data are represented as mean percentages of mClover-positive cells over the iRFP-positive population from five independent experiments (total n > 100 iRFP-positive cells) and reported in a scatter dot plot representing SEM, and a classical one-way Anova test was performed.

## GFP-Trap pulldown

HEK293T were plated to 80% confluency and transfected with YFP-CTL, YFP-PALB2 or YFP-PALB2-146AAAA vector using Lipofectamine RNAiMAX Reagent (Thermo Fisher Scientific). 24 hr after, cells were irradiated with 5 Gy and processed for GFP-Trap Pulldown 2 hr post-irradiation. Briefly, cells were washed twice with ice cold PBS and lysed in lysis buffer (50 mM Tris-HCl, pH 7.5, 150 mM NaCl, 0,5% NP40, containing PMSF, Aprotinin, Leupeptin, NaF, and $Na_2VO_4$). Lysates were soni-cated (three times 10 s at 30% amplitude on ice) and centrifuged at 16000 g, for 30 min at 4°C. Protein lysates (3 mg) were incubated with GFP-Trap beads (Chromotek) for 1 hr at 4°C. Beads were washed three times with lysis buffer (without NP40) and bound proteins were resuspended in 35 μL of Laemmli SDS-sample buffer and heated to 95°C for 10 min.

Samples were run on NuPAGE 4–12% bis-Tris Protein Gels (Invitrogen) in NuPAGE MOPS SDS Running Buffer according to the manufacturer's protocol and transferred to Nitrocellulose membrane (Amersham) using XCell II Blot Module (Invitrogen) in 20% methanol transfer buffer. Immunoblots were performed using the following antibodies: anti-RAD51(14B4) (Novus Biologicals, #NB100-148) and anti-GFP (Roche, #11814460001).

## Plasmids and siRNA

peYFP-C1-PALB2 was modified to be resistant to PALB2 siRNA by Q5 Site-Directed Mutagenesis Kit (NEB, E0554) using primers JYM3892/3893 (*Supplementary file 1E*). The resulting siRNA-resistant construct was then used as a template to generate the mutant construct YFP-PALB2 146AAAA with the primers JYM3909/JYM3910. Flag-tagged PALB2 146AAAA mutant was also obtained via site-directed mutagenesis on pcDNA3-Flag PALB2 (*Pauty et al., 2017*).

## Cell lines

U2OS (HTB-96) and HEK293T (CRL-11268) were purchased from ATCC and HeLa were authenticated using Short Tandem Repeat (STR) analysis by ATCC services (100% match). All the cells lines used were uninfected with Mycoplasma, as routinely verified using the e-Myco Mycoplasma kit from FroggaBio.

## Acknowledgements

We are grateful to the members of Korolev lab including Ian Miller for help with cloning, Lakshmi Kanikkannan and Jennifer Redington for help with protein purification and DNA-binding assays. We are grateful to Drs Alessandro Vindigni and Joel Eissenberg for the manuscript evaluation and discussions. We thank Amélie Rodrigue and Yan Coulombe from the Masson lab for technical support. J-YM is a FRQS Chair in genome stability. The research was supported by Siteman Cancer Center (SCC) and the Foundation for Barnes-Jewish Hospital Siteman Investment Program (SIP) [Pre-R01 award to SK]; the Canadian Institutes of Health Research (GD); and a Canadian Institutes of Health Research Foundation grant to J-YM Funding for open access charge: National Institute of Health.

## Additional information

### Funding

| Funder | Author |
|---|---|
| Siteman Cancer Center | Sergey Korolev |
| Foundation for Barnes-Jewish Hospital | Sergey Korolev |
| Canadian Institutes of Health Research | Jean-Yves Masson |

The funders had no role in study design, data collection and interpretation, or the decision to submit the work for publication.

### Author contributions

Jaigeeth Deveryshetty, Data curation, Validation, Investigation, Writing—original draft; Thibaut Peterlini, Data curation, Formal analysis, Validation, Investigation, Visualization; Mikhail Ryzhikov, Formal analysis, Validation, Investigation, Methodology; Nadine Brahiti, Formal analysis, Validation, Investigation; Graham Dellaire, Resources, Methodology; Jean-Yves Masson, Supervision, Funding acquisition, Validation, Methodology, Writing—original draft; Sergey Korolev, Data curation, Supervision, Funding acquisition, Validation, Methodology, Writing—original draft, Project administration, Writing—review and editing

### Author ORCIDs

Graham Dellaire https://orcid.org/0000-0002-3466-6316
Sergey Korolev https://orcid.org/0000-0001-9313-7126

### Decision letter and Author response

Decision letter https://doi.org/10.7554/eLife.44063.032
Author response https://doi.org/10.7554/eLife.44063.033

## Additional files

### Supplementary files

• Supplementary file 1. Supplementary tables including. (A) DNA binding parameters of N-DBD mutants, and DNA/RNA sequences for (B) primers for DNA binding site mutagenesis, (C) substrates

for DNA binding assay, (D) Substrates for strand exchange activity and FRET assays, (E) siRNA resistance primers.

DOI: https://doi.org/10.7554/eLife.44063.027

• Transparent reporting form

DOI: https://doi.org/10.7554/eLife.44063.028

## Data availability

All data generated or analyzed during this study are included in the manuscript and supporting files.

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
