## [Decision Letter]

Thank you for submitting your article "Novel strand exchange activity of the human PALB2 DNA Binding Domain and its critical role for DNA repair in cells" for consideration by *eLife*. Your article has been reviewed by three peer reviewers, including Maria Spies as the Reviewing Editor and Reviewer #1, and the evaluation has been overseen by Cynthia Wolberger as the Senior Editor. The following individual involved in review of your submission has also agreed to reveal his identity: Mark Simon Dillingham (Reviewer #2).

The reviewers have discussed the reviews with one another and the Reviewing Editor has drafted this decision to help you prepare a revised submission.

Summary:

Your manuscript has been reviewed by three experts, who found your findings potentially interesting and important, especially the DNA strand exchange activity of the DBD of PALB2. This is an important observation because it suggests that the PALB2's coordination of the BRCA2, RAD51 etc. in DSB repair is not a mere hub function. Biochemical studies being paralleled by the observations in the cell are a definite strength of the paper. The reviewers. However, have identified a number of problems with the work, which need to be resolved prior to possible consideration.

Essential revisions:

Significant issues that will require additional experiments:

1) The reported DNA (and RNA) strand exchange activity of PALB2 DBD is the main novelty of the paper. This said, it is important to distinguish the ability of PALB2 to exchange short, protein-free segments of nucleic acids from the DNA strand exchange reaction that is expected to occur at the resected DSBs and which involves longer stretches of RPA-coated ssDNA. The authors need to make this distinction clear in the Discussion. Considering the importance of PALB2-DNA interaction for DSB repair in the cell-based studies, it would be informative to test the T1 effect on the DNA strand exchange reaction involving the plasmid length substrates, such as plasmid lengths circular ssDNA and linear duplex, or a D-loop assay with supercoiled plasmid-length dsDNA. The D-loop experiment will also allow you to compare the DBD with the full-length PALB2 (you can compare these observations side by side with the published data if you don't have access to a full-length PALB2).

2) It is important to determine whether the DNA strand exchange activity, which is observed to be ATP dependent, or if PALB2 DBD acts more like RAD52. The absence of the nuclease activity in the DBD preparation should be also demonstrated as even a trace amount of nuclease activity may alter the results of the FRET-based experiments.

3) There is no evidence that the mutant proteins are properly folded or that any other intrinsic activity is maintained at wild type levels. Without further validation it is not possible to conclude that the residues directly affect DNA binding (although it is likely), nor that these mutations uncouple DNA binding from other activities of PALB2 (which is an important underlying assumption for the in vivo experiments). The authors should demonstrate that their mutant proteins are properly folded. CD spectra would be the best if the DBD possesses discernable secondary structure. The effect of the mutations on the oligomeric state of the DBD should also be probed.

4) Figure 6: The choice of the FRET substrate with the dyes separated by 70 nucleotides may lead to an erroneous conclusion about the wrapping vs. extension of the ssDNA bound by DBD and RAD51/DBD complex. These experiments should be repeated with the substrate with the dyes positioned within the linear FRET range. It is quite unexpected for a 70-mer Cy3/Cy5-labeled substrate to exhibit an increase in the FRET signal upon RAD51 binding. This could be due to how the FRET is calculated and the fact that RAD51 enhances Cy3 fluorescence upon binding. As presented, an apparent FRET is calculated without taking into account the leakage between the channels and the difference in quantum yield of Cy3 and Cy5. While the data still support the idea of the co-complex involving RAD51 and PALB2, there is no way to deduce the DNA configuration from these data. To argue about the ssDNA configuration (ssDNA extended by RAD51 and wrapped by T1), you should repeat the experiments with a DNA substrate where the dyes are separated by no more than 25 – 30 nt. This range is more compatible with FRET. Also, the SSB-like wrapping of the ssDNA by the T1 fragment can be revealed using the substrate with the internal placement of the dyes. For the terminally-placed dyes, one can formally argue that the FRET increase is due to the bridging of the two ends.

Significant issues that will require additional analysis/explanations

1) Quantification of the rates and the reaction extents in Figures 5 and 7 would be highly informative.

2) The reviewers felt that the DNA binding data are neither presented well nor sufficiently discussed. The authors should provide the equation used to quantify DNA binding. Considering the weak binding and potential cooperativity, it would be the most appropriate if an implicit Hill equation, which takes into account the difference between total and free concentration of the protein, were used. It is unclear, however, whether this was the case. The assumption of the oligomeric state used in the data fitting needs to be explicitly stated. Furthermore, the authors should discuss the evidence for cooperativity in DNA binding (if any) and report the fitting parameters.

3) The proposed oligomeric state of the DBD should be discussed in light of the previous work (Buisson et al) that reported a dimer for the full-length PALB2. The K-A mutant binding isotherms plateau at different anisotropy values compared to the control. It is possible that the mutant protein is adopting different oligomeric states. A size exclusion chromatogram of the mutant proteins would address this concern.

4) The protein preps of 573 or 5753-146 AAAA appears to contain degradation or contaminating proteins. How was the protein quantified? This is important consideration for the binding assays.

Suggestions for improving readability:

1) The paper would significantly benefit from editing. For example, the Introduction reads like a long list of facts and it is unclear how these facts go together. Stating up front the rationale behind the experiment and results would make the paper much easier to follow. For example, why were "Qualitative measurement of PALB2 interaction with ss- and dsDNA oligonucleotides of.…" analyzed? The authors should state the hypothesis.

2) Important previous characterization data were omitted from the Introduction. For example, characterization of PALB2 from Buisson R. et al., 2010:

a) PALB2 binds DNA substrates via two regions (explicitly list those domains).

b) Full length PALB2 does not stimulate strand exchange reactions.

c) Full length PALB2 promotes the strand RAD51-mediated strand exchange reaction.

3) The rational for testing the inverse and RNA strand exchange activities should be explicitly discussed.

---

## [Author Response]

Essential revisions:Significant issues that will require additional experiments:1) The reported DNA (and RNA) strand exchange activity of PALB2 DBD is the main novelty of the paper. This said, it is important to distinguish the ability of PALB2 to exchange short, protein-free segments of nucleic acids from the DNA strand exchange reaction that is expected to occur at the resected DSBs and which involves longer stretches of RPA-coated ssDNA. The authors need to make this distinction clear in the Discussion. Considering the importance of PALB2-DNA interaction for DSB repair in the cell-based studies, it would be informative to test the T1 effect on the DNA strand exchange reaction involving the plasmid length substrates, such as plasmid lengths circular ssDNA and linear duplex, or a D-loop assay with supercoiled plasmid-length dsDNA. The D-loop experiment will also allow you to compare the DBD with the full-length PALB2 (you can compare these observations side by side with the published data if you don't have access to a full-length PALB2).

A) We performed a D-loop assay with ssDNA and supercoiled dsDNA plasmid. The PALB2 N-DBD fragment does not support D-loop formation with a supercoiled plasmid. Results are shown in Figure 4—figure supplement 4. We added discussion of differences in D-loop formation between N-DBD and full length PALB2.

B) We tested RPA in strand exchange assay (new Figure 6). 0.5 μM RPA significantly inhibits N-DBD- and N-DBD/RAD51-mediated strand exchange.

2) It is important to determine whether the DNA strand exchange activity, which is observed to be ATP dependent, or if PALB2 DBD acts more like RAD52.

The PALB2 strand exchange is ATP-independent. All experiments without RAD51 were performed without ATP. We now clarify this in the main text and Materials and methods.

The absence of the nuclease activity in the DBD preparation should be also demonstrated as even a trace amount of nuclease activity may alter the results of the FRET-based experiments.

There is no measurable DNA degradation upon one hour incubation at 37°C with PALB2 in gel shifts (Figure 4—figure supplement 2). We have tested the stability of DNA substrates multiple times. We also did not observe any change in fluorescence when proteins were incubated with single-labeled Cy3 or Cy5 (Author response image 1).

**Author response image 1. respfig1:** DNA and T1 were incubated at 37C for 30 min and treated with proteinase K for 20 min and ran on Native page gel at 60 V for 30 min. DNA without protein were treated exactly the same and compared on rhe gel.

3) There is no evidence that the mutant proteins are properly folded or that any other intrinsic activity is maintained at wild type levels. Without further validation it is not possible to conclude that the residues directly affect DNA binding (although it is likely), nor that these mutations uncouple DNA binding from other activities of PALB2 (which is an important underlying assumption for the in vivo experiments). The authors should demonstrate that their mutant proteins are properly folded. CD spectra would be the best if the DBD possesses discernable secondary structure. The effect of the mutations on the oligomeric state of the DBD should also be probed.

The mutant protein has similar solubility to that of the wild type during all purification steps. It does form tetramers (new Figure 8—figure supplement 2A). It interacts with RAD51 (new Figure 3—figure supplement 3). Therefore, protein misfolding is highly unlikely.

We do not have access to CD and cannot rule out minor conformational changes. Neither T1 nor 573 fragments were suitable for thermal denaturation assays either with Sypro Orange kit or using DSC Microcalorimeter (readily precipitated).

Overall, mutation of several basic residues to alanine is unlikely to have a significant effect on the folding of 1183 aa protein in cell. The effect is negligible compared to deletions of large domain commonly used in cellular studies of PALB2 and other proteins.

4) Figure 6: The choice of the FRET substrate with the dyes separated by 70 nucleotides may lead to an erroneous conclusion about the wrapping vs. extension of the ssDNA bound by DBD and RAD51/DBD complex. These experiments should be repeated with the substrate with the dyes positioned within the linear FRET range. It is quite unexpected for a 70-mer Cy3/Cy5-labeled substrate to exhibit an increase in the FRET signal upon RAD51 binding. This could be due to how the FRET is calculated and the fact that RAD51 enhances Cy3 fluorescence upon binding. As presented, an apparent FRET is calculated without taking into account the leakage between the channels and the difference in quantum yield of Cy3 and Cy5. While the data still support the idea of the co-complex involving RAD51 and PALB2, there is no way to deduce the DNA configuration from these data. To argue about the ssDNA configuration (ssDNA extended by RAD51 and wrapped by T1), you should repeat the experiments with a DNA substrate where the dyes are separated by no more than 25 – 30 nt. This range is more compatible with FRET. Also, the SSB-like wrapping of the ssDNA by the T1 fragment can be revealed using the substrate with the internal placement of the dyes. For the terminally-placed dyes, one can formally argue that the FRET increase is due to the bridging of the two ends.

Note, that the increase of FRET upon RAD51 binding was observed in the absence of ATP, and FRET decreases when ATP is added. Nevertheless, we agree that this experiment is inconclusive and requires additional controls. We decided to remove this experiment (former Figure 6) from main text.

In the revised version, we show only FRET experiments with 40- and 70-nt ssDNA in Figure 8—figure supplement 2. The leakage between channels was measured and incorporated into the FRET estimate as described now in Materials and methods.

Significant issues that will require additional analysis/explanations1) Quantification of the rates and the reaction extents in Figures 5 and 7 would be highly informative. Figure 5D may be represented with units/sec for the initial phase.

We found that for current data, the rates do not provide any additional information. We calculated initial rates for the strand exchange experiments and the rates for the new Figure 7 (Author response image 2). It will not provide valuable information for comparison with similar activity of RAD52 (Mazina et al., 2017), where all data are reported in product% and not with rate values. However, we did add comparisons with apparent rates in our and the referred paper in the Discussion.

**Author response image 2. respfig2:** Table with initial rates (bottom) calculated for reactions in Figure 7.

We agree with the reviewers that proper quantification is necessary for mechanistic insight into PALB2 DNA-binding and strand-exchange. However, comprehensive mechanistic studies of this system will require a significant number of additional experiments and are beyond the scope of this manuscript.

2) The reviewers felt that the DNA binding data are neither presented well nor sufficiently discussed. The authors should provide the equation used to quantify DNA binding. Considering the weak binding and potential cooperativity, it would be the most appropriate if an implicit Hill equation, which takes into account the difference between total and free concentration of the protein, were used. It is unclear, however, whether this was the case.

Thank you for pointing to this important gap in initial presentation. We described DNA binding in more details in Materials and methods, included Hill coefficient measurements and tables with binding parameters. Hill coefficient was closure to 1, suggesting noncooperative binding, except T1 binding to ds49, which may be an artifact of the short domain, as it is not the case for 573 fragment.

The effect of protein depletion is not encountered (accordingly to recommendations for FP assay in Prism). We referred it as an apparent equilibrium dissociation constant. We have the changed main text and Materials and methods accordingly.

The assumption of the oligomeric state used in the data fitting needs to be explicitly stated. Furthermore, the authors should discuss the evidence for cooperativity in DNA binding (if any) and report the fitting parameters.

We did not account for oligomeric state in DNA-binding fitting. As mentioned above, we do not observe significant cooperativity. Instead, we observed higher affinity towards longer ssDNA substrates, suggesting interaction of long ssDNA with multiple sites. We speculate that multiple binding sites can be provided by an oligomeric T1 structure observed in SEC experiments. This situation resembles ssDNA binding by SSB. We are currently isolating mutants affecting T1 oligomerization, but this will take additional time to study.

3) The proposed oligomeric state of the DBD should be discussed in light of the previous work (Buisson et al.) that reported a dimer for the full-length PALB2.

We have added corresponding discussion.

The K-A mutant binding isotherms plateau at different anisotropy values compared to the control. It is possible that the mutant protein is adopting different oligomeric states. A size exclusion chromatogram of the mutant proteins would address this concern.

The mutant also forms tetramer. Due to weak binding, anisotropy isotherms do not reach plateaus in case of mutants, explaining their low anisotropy values. Overall, the T1 truncation and its mutants are relatively unstable under the binding conditions. We are trying to isolate a more stable truncation suitable for future quantitative mechanistic studies of PALB2 DNA interaction and strand annealing.

4) The protein preps of 573 or 5753-146 AAAA appears to contain degradation or contaminating proteins. How was the protein quantified? This is important consideration for the binding assays.

We present data for newly purified proteins. While modified purification methods with an additional chromatography step yield cleaner products, we cannot completely eliminate minor contamination by a likely degradation product. New preps have similar binding parameters. Such minor contamination should not affect comparative analysis of wild type and mutant proteins identically purified.

Suggestions for improving readability:1) The paper would significantly benefit from editing. For example, the Introduction reads like a long list of facts and it is unclear how these facts go together. Stating up front the rationale behind the experiment and results would make the paper much easier to follow. For example, why were "Qualitative measurement of PALB2 interaction with ss- and dsDNA oligonucleotides of.…" analyzed? The authors should state the hypothesis.

This is an important point and we have tried to improve our presentation. We mentioned controversial results from previous publications and added additional description of the significance of DNA binding for the function of recombination mediator proteins.

2) Important previous characterization data were omitted from the Introduction. For example, characterization of PALB2 from Buisson R. et al., 2010:a) PALB2 binds DNA substrates via two regions (explicitly list those domains).

Added.

b) Full length PALB2 does not stimulate strand exchange reactions.

We added discussion of this fact (subsection “PALB2 stimulates RAD51-mediated strand exchange and promotes a similar reaction without RAD51”).

c) Full length PALB2 promotes the strand RAD51-mediated strand exchange reaction.

We added discussion of this fact (subsection “PALB2 stimulates RAD51-mediated strand exchange and promotes a similar reaction without RAD51”).

3) The rational for testing the inverse and RNA strand exchange activities should be explicitly discussed.

See subsection “PALB2 stimulates an inverse strand exchange and can use an RNA substrate”.